# Sampling density and date along with species selection influence spatial representation of tree-ring reconstructions

Justin T. Maxwell[1,2], Grant L. Harley[3], Trevis J. Matheus[4], Brandon M. Strange[5], Kayla Van Aken[6], Tsun Fung Au[1], and Joshua C. Bregy[1,7]

1 Department of Geography, Indiana University
2 Harvard Forest, Harvard University
3 Department of Geography and Geological Sciences, University of Idaho
4 Department of Geography and the Environment, California State University, Fullerton
5 School of Natural Resources and the Environment, University of Arizona
6 School of Biological, Environmental and Earth Sciences, University of Southern Mississippi
7 Department of Earth and Atmospheric Sciences, Indiana University

*Correspondence to*: Justin T. Maxwell (maxweljt@indiana.edu)

**Abstract.** Our understanding of the natural variability of hydroclimate before the instrumental period (*ca.* 1900 in the United States; US) is largely dependent on tree-ring-based reconstructions. Large-scale soil moisture reconstructions from a network of tree-ring chronologies have greatly improved our understanding of the spatial and temporal variability in hydroclimate conditions, particularly extremes of both drought and pluvial (wet) events. However, certain regions within these large-scale, network reconstructions in the US are modeled by few tree-ring chronologies. Further, many of the chronologies currently publicly available on the International Tree-Ring Data Bank (ITRDB) were collected in the 1980s and 1990s, thus our understanding of the sensitivity of radial growth to soil moisture in the US is based on a period that experienced multiple extremely severe droughts and neglects the impacts of recent, rapid global change. In this study, we expanded the tree-ring network of the Ohio River Valley in the US, a region with sparse coverage. We used a total of 72 chronologies across 15 species to examine how increasing the density of the tree-ring network influences the representation of reconstructing the Palmer Meteorological Drought Index (PMDI). Further, we tested how the sampling date and therefore the calibration period influenced the reconstruction models by creating reconstructions that ended in the year 1980 and compared them to reconstructions ending in 2010 from the same chronologies. We found that increasing the density of the tree-ring network resulted in reconstructed values that better matched the spatial variability of instrumentally-recorded droughts, and to a lesser extent, pluvials. By extending the calibration period to 2010 compared to 1980, the sensitivity of tree rings to PMDI decreased in the southern portion of our region where severe drought conditions have been absent over recent decades. We emphasize the need of building a high-density tree-ring network to better represent the spatial variability of past droughts and pluvials. Further, chronologies on the ITRDB need updating regularly to better understand how the sensitivity of tree rings to climate may vary through time.

Keywords: Drought, Pluvial, Midwest United States, Dendrochronology, Palmer Meteorological Drought Index

# 1 Introduction

Understanding the mechanisms that drive climate variability, particularly before the modern instrumental record (*ca.* 1900 in the United States; US), depends on proxy-based reconstructions of climate. Precisely-dated tree-ring chronologies are one of the primary proxies that can reconstruct inter-annual climate variability over recent centuries to millennia (Fritts, 1976). Tree rings provide robust historical and prehistorical context for droughts and pluvials (wet periods) captured in the instrumental record throughout the mid-latitudes (*e.g.,* Stahle and Cleaveland 1994; Woodhouse and Overpeck, 1998; Cook *et al.*, 2010; Fang *et al.* 2010; Chen *et al.* 2013; Pederson *et al.*, 2013; Güner *et al.*, 2017; Oliver *et al.* 2019; Morales *et al.* 2020). Most of our understanding of past drought severity and variability in North America is the result of the North American Drought Atlas (NADA; Cook *et al.*, 1999). The NADA comprises a network of tree-ring chronologies across North America from the International Tree-Ring Data Bank (ITRDB; https://www.ncdc.noaa.gov/data-access/paleoclimatology-data/datasets/tree-ring), creating a 2° x 3° reconstruction of summer (average of June, July, and August; JJA) Palmer Drought Severity Index values (Palmer, 1965). The NADA produced multiple centuries of spatial drought variability, providing essential context for extreme soil-moisture conditions witnessed in the most recent centuries. More recently, the Living Blended Drought Atlas (LBDA; Cook *et al.*, 2010) updated the NADA using additional tree-ring chronologies from the ITRDB and higher spatial-resolution climate data to calibrate models, creating a 0.5° x 0.5° reconstruction of the Palmer Meteorological Drought Index (PMDI; Palmer, 1965).

While the NADA and LBDA have provided invaluable information of past droughts and pluvials in North America, they were generated to compare large, sub-continental events. The reconstruction at each grid cell uses tree-ring data that are within a 450-km radius of that grid point. By pulling from such a wide range of predictors, the NADA and LBDA models excel at representing large-scale hydroclimate variablity as they tend to average out smaller scale features. However, these drought atlases may not represent local conditions in areas with sparse coverage of tree-ring chronologies, such as certain regions of the midwestern US (Maxwell and Harley, 2017; Strange *et al.*, 2019). The tree-ring chronologies from the ITRDB can have biases related to tree species used and the spatial density of the tree-ring network

(Zhao *et al*., 2019; Coulthard *et al.,* 2020). When collecting tree-ring data for the purpose of reconstructing climate, the general goal is to target long-lived species that are sensitive to the climate variable to be reconstructed while also maximizing the length of the reconstruction. However, inclusion of multiple species in a reconstruction can improve model performance and skill (Pederson *et al.*, 2001; Frank and Esper, 2005; Cook and Pederson, 2011; Maxwell *et al.*, 2011; Pederson *et al.*, 2012; Maxwell *et al.*, 2015). In the US, the ITRDB has excellent spatial replication in certain regions, such as the American Southwest, but other regions are poorly represented, such as the Ohio River Valley (ORV; Zhao *et al.*, 2019). Due to changes in the density of the tree-ring network of the ITRDB and the use of a large radius (450 km) to reconstruct drought for the LBDA, soil moisture variability at local scales is potentially absent in areas that are underrepresented in the tree-ring network. Further, many of the chronologies that are available on the ITRDB were collected in the 1980s and have not been updated, limiting the range of climatic conditions to calibrate reconstruction models (Larson *et al.* 2013; Zhao *et al.*, 2019).

The wealth of climate information derived from tree rings is based on the key assertion that their physiological development is related to specific climatic conditions. An explicit relationship between climate and tree growth can be estimated during the instrumental period. Yet, developing a reconstruction assumes that this climate-tree-growth relationship is stationary over time. This assumption was generally true in the early development of the field of dendrochronology (Fritts, 1976). However, as human activities drive the Earth's climate system into historically unprecedented, and potentially non-stationary and non-analogous conditions (Milly *et al.,* 2008), exceptions to this assumption have emerged. Changes in the drought signal recorded by tree rings have been established only recently in the eastern US (Larson *et al.,* 2013; Maxwell *et al.,* 2015, 2016, 2019; Helcoski *et al.,* 2019), making an investigation of its causes essential to ensuring the interpretability of tree-ring-based hydroclimate reconstructions. Of these recent studies, Maxwell *et al.* (2016) provided the first documentation of an apparent deteriorating relationship between radial tree growth and summer soil moisture that is not accompanied by an increase in signal strength during another season. The declining relationship—referred to as the "Fading Drought Signal"— was consistent across multiple species and sites within the Central Hardwoods Forest region of the midwestern US. However, Maxwell *et al.* (2019) found that *Acer* (maple) species had a stable relationship, implying that including species from this genus in reconstructions could improve model performance. In

this paper, we test the hypothesis that increasing the spatial density of the tree-ring network results in reconstructions that better replicate the local variation of the instrumental data despite a fading drought signal. We also examine if the period in which the tree-ring data is calibrated with climate data influences the climate reconstruction. Using the new, dense tree-ring network of the ORV, we calibrate the reconstruction with recent (post-1980) radial growth and climate data and compare to reconstructions generated using data only from pre-1980. We test the hypothesis that including recent data could reduce the amount of variance explained in tree-ring reconstruction of soil moisture in the ORV.

## 2 Methods

### 2.1 *Living Blended Drought Atlas*

For the LBDA, Cook *et al.* (2010) created a gridded instrumental dataset of PMDI to calibrate tree-ring reconstruction models. The instrumental data were created using observations for temperature and precipitation from over 5,000 and 7,000 weather stations, respectively, which were spatially interpolated with a trivariate thin-plate spline in the ANUSPLIN program (Hutchinson, 1995). Cook *et al.* (2010) derived the reconstructions by gathering standardized tree-ring chronologies within 450 km of each instrumental grid point center. However, because the LBDA was developed across North America, Cook *et al.* (2010) used a dynamic search radius, with the requirement of having a minimum of five chronologies as possible predictors; so in certain regions, the radius was larger than 450 km. Therefore, in sparsely covered areas such as the ORV, the actual search radius for the LBDA could be larger than 450 km. Chronologies that were significantly ($p < 0.05$)correlated with PMDI were retained and used in a principal component analysis (PCA). The resulting principal components (PCs) that had eigenvalues greater than one were then used as predictors in the reconstruction model. For the LBDA, we gathered both the instrumental and reconstructed 0.5° x 0.5° gridded PMDI data for the ORV region (Figure 1)

from the National Oceanic and Atmospheric Administration, National Center for Environmental Information (https://www.ncdc.noaa.gov/paleo-search/study/19119; Cook *et al.,* 2010).

## 2.2 *Ohio River Valley Tree-Ring Network*

To examine how the density of the tree-ring network could impact the reconstruction, we gathered recently published chronologies and collected new chronologies across the ORV to fill the spatial gaps of the ITRDB (Figure 1; Supplemental Table 1). For the new chronologies, we either 1) updated existing chronologies from the ITRDB; 2) sampled new co-occurring species at an ITRDB site; or 3) created new chronologies from previously unsampled sites. For this study, we used a total of 72 chronologies across 15 species. Of these chronologies, 37 were published, three were newly updated ITRDB records, and 32 were new collections (Figure 1; Supplemental Table 1). For the new ($n = 32$) and updated ($n = 3$) chronologies, we used standard field methods to target at least ten old growth trees for each species using morphological characteristics (Pederson, 2010). We used a hand-held 4.3-mm-diameter increment borer to extract two samples from each tree at breast height, from opposite sides of the tree (Stokes and Smiley, 1968). All newly collected samples were mounted and sanded with progressively finer sandpaper to reveal ring structure. We used the list method to visually crossdate all samples (Yamaguchi, 1991), and then the program COFECHA (Holmes, 1983) to statistically verify the crossdating. For the three updated chronologies, we crossdated the new sampled series with those previously sampled and available through the ITRDB.

## 2.3 *Detrending Tree-Ring Series*

For all chronologies, we removed both age-related growth trends and non-climatic influences of tree growth (*e.g.*, forest dynamics or insect outbreaks) by using signal-free standardization (Melvin and Briffa, 2008) with a two-thirds smoothing spline applied to each measured series (Cook and Peters, 1981). To ensure we achieved the desired spline flexibility of the two-thirds spline in the standardization, we used the approximation suggested by Bussberg *et al.* (2020) and used an 83% spline to account for endpoint adjustments. We stabilized the variance of the standardized chronologies using the data-adaptive power transformation (Cook and Peters, 1997). Signal-free standardization can reduce "trend distortion"

problems near the ends of the record (Melvin and Briffa, 2008). We trimmed each chronology to remove the portion of the record where low sample depth inflated the variance in standardized growth using an expressed population signal (EPS) value of 0.80 (Wigley *et al.*, 1984).

### 2.4 *Point-by-Point Regression*

We replicated the point-by-point regression procedure for the LBDA in Cook *et al.* (2010) and described in Cook *et al.* (1999) for the ORV tree-ring network. We developed a network of 0.5° x 0.5° grid points reconstructions ($n$ = 181) across the ORV region, defined as 37.75–42.25° N, 82.25–90.75° W (Figure 1). Similar to the LBDA, we produced PMDI reconstructions at each grid point by first screening standardized tree-ring chronologies through correlation analysis with PMDI from 1895 to 2010, where only the chronologies with significant ($p < 0.05$) correlations were retained. Both the tree-ring chronologies and the climate data were prewhitened during this screening procedure to remove the influence of short-term autocorrelation.

To examine how increasing the density of the tree-ring network influences the reconstruction, we gathered tree-ring chronologies within a 250-km radius from the center of each grid point instead of the 450-km minimum radius used for LBDA. For the ORV gridded reconstructions, the use of a 250-km radius ensured that each gridded reconstruction could have at least five chronologies as possible predictors (Supplemental Figure 1). For each grid point, we built a reconstruction model by taking the screened standardized chronologies and using both the current year ($t$) and the following year ($t$+1) as possible predictors due to current year climate conditions impacting growth both during the current and the proceeding year, which doubled the number of predictors. We then took all the $t$ and the $t$+1 chronologies that passed the screening and conducted a PCA. Per the Kaiser-Guttman rule (Guttman, 1954, Kaiser, 1960), we then used the PCs with eigenvalues greater than one as predictors in a regression model to predict mean JJA PMDI. To ensure that our ORV reconstruction was comparable to the LBDA, we added the autocorrelation of the instrumental data back into the final tree-ring reconstructions of PMDI as done for the NADA and LBDA.

We used Pearson's correlation to compare the reconstructed PMDI values from the LBDA to the ORV reconstruction at each grid point. We further chose well-known drought and pluvial years in the instrumental period to examine how the ORV and LBDA compared spatially. Specifically, we examined the droughts of 1988, 1954, 1936, 1816, and 1774 and the pluvials of 1945–1951, 1882–1883, and 1811 (Trenberth *et al.* 1988; Stambaugh *et al.* 2011; Heim 2017) . To compare the reconstructions with the instrumental data, we calculated the mean absolute error for each extreme event. We also correlated the instrumental PMDI at each grid-point to every other grid-point and then examined those correlations as a function of distance. Similarly, the reconstructed PMDI values were correlated for each grid-point for the ORV and LBDA and compared across distance. To examine the species contribution to the overall ORV reconstruction, we gathered the correlation of each species chronology to the PMDI for each grid reconstruction that the given species were included.

## 2.5 *Droughts and Pluvials*

To determine if the ORV and LBDA reconstructions had differences in the amount of extreme hydroclimatic conditions, we calculated the number of years in each gridded reconstruction that had a JJA PMDI value of $\geq 2.0$ or $\leq -2.0$ to represent at least moderately wet and dry conditions, respectively. We further examined how the volatility in extreme conditions compared between the two reconstructions by calculating "flips" from one extreme to the other in consecutive years (Loecke *et al.* 2017; Oliver *et al.,* 2019; Harley *et al*., 2020). We specifically used an index developed by Loecke *et al.* (2017) to quantify large "whiplashes" (termed flips here) interannually. The flip index is defined as:

$i = \text{PMDI} (t + 1) – t / \text{PMDI} (t + (t + 1))$

where the index (i) equals the PMDI value of a given year (t) subtracted from the PMDI value of the following year (t +1), divided by the sum of the PMDI values over the two-year period (t+(t+1)). Positive index values indicate that conditions shifted from dry to wet over the two-year period. Similarly, negative values represent a shift from wet to dry conditions. We used an index value > 75[th] percentile to define an abnormally wet period and < 25[th] percentile an extremely dry period. We then calculated wet flip events as years that were abnormally dry followed directly by extreme wet years. Dry flips were calculated as abnormally wet years followed by extreme drought years. Lastly, we

summed the wet and dry flips to calculate the total flips. These flips were calculated for each grid point in the ORV reconstruction where sample depth was determined by an EPS value of 0.80 to reproduce the variance in the instrumental data (Wigley *et al.,* 1984). We limited the calculation of flips to the period 1658–2005, which was the common period of overlap between the longest gridded ORV reconstruction and the LBDA.

### 2.6 *Model Validation Comparisons*

To examine the temporal stability of the relationship between tree growth and PMDI, we followed the same validation procedures used for the LBDA (Cook *et al.*, 2010). We used the early half of the common period (1901–1955) to calibrate a model between tree growth and PMDI to validate the late half (1956–2010). We used two tests of fit, the reduction of error statistic (RE) and the coefficient of efficiency (CE; Fritts, 1976; Cook *et al.,* 1999), to validate our calibration models. RE and CE both range from $-\infty$ to $+1$, with positive values indicating robust predictive skill. However, RE is compared to the mean of the instrumental data, while CE relies on the verification period mean and therefore is a more conservative verification metric. We then compared the variance explained ($R^2$), RE, and CE values between the LBDA and the ORV PMDI reconstructions for each grid point. We also mapped the gridded reconstructed PMDI values from extreme years in the observation period and well-known years in the historical record for both the LBDA and the ORV reconstructions to provide examples of the spatial differences between the two reconstructions.

To examine how validation statistics may change based on when the trees were sampled, we created a second ORV reconstruction where the most recent year was 1980. This year was chosen because several chronologies available on the ITRDB were sampled in the 1980s, and this marked the beginning of a weakening relationship between radial growth and soil moisture in this region (Maxwell *et al.*, 2016). We used the same validation process described above except the early period was from 1901 to 1940 and the late period was from 1941 to 1980. We then calculated the difference between the 1980 and the 2010 reconstruction for $R^2$, RE, and CE values for each grid point.

# 3 Results

## 3.1 ORV vs. LBDA

Our first comparisons of chronologies distributed for the LBDA and ORV networks revealed broad spatial discrepancies. PMDI point-by-point regressions for the LBDA included 20 chronologies from six species over the study region, whereas the ORV network included 72 chronologies from 15 tree species. Not only is the spatial density of sites sparser for the LBDA network, but it mostly only included single-chronology sites, whereas 18 of the sites included in the ORV are multiple-species sites (2–6 co-occurring species) (Figure 1A, B). Although site coverage is sparse for both networks along the west-central, northwest, and southeast sectors, the ORV network included major spatial coverage improvements in other sectors (Figure 1). Particularly, the ORV increased spatial coverage in south-central Indiana where many of the sites included four to six co-occurring species chronologies ($n = 27$ total chronologies). The PMDI reconstructions from the ORV network and the LBDA demonstrated strong and positive correlations, with $r$-values ranging from 0.50 to 0.90 (Figure 2). These correlations were calculated for the period of overlap between the two gridded reconstructions, 1830–2005 C.E. The highest correlations were found along the western portion of the gridded region, while the lowest agreement was found in the southeast (Figure 2).

The ORV reconstructions were shorter in length (maximum of 343 years) compared to the LBDA reconstructions (maximum of 1,645 years) due to needing numerous old chronologies to load into each grid reconstruction. While this is true for the LBDA, having a larger search radius allows a longer chronology to be included in many gridded reconstructions. A smaller search radius for chronology inclusion requires a denser network of longer chronologies to reach a similar length as the LBDA. Secondly, we focused on increasing the spatial density of the network, which resulted in sampling younger sites (*e.g.,* the earliest years are in the early to late 19$^{th}$ century). While the ORV reconstructions were

shorter, comparing certain well-known extreme climatic years during the period of the overlap between the LBDA show some important differences.

## 3.2 *ORV and LBDA Extreme Year Comparisons*

We chose a series of well-known drought and pluvial years (events) to compare the reconstructions between ORV and LBDA. In general, the increased spatial density of tree-ring chronologies used in the ORV reconstruction displayed more local variation in the reconstructions of extreme climatic events (Figure 3). However, in a few examples, such as 1774 and 1816, the spatial pattern of where extreme drought was located changed between the two reconstructions (Figure 3). Using extreme events in the observed record (three droughts and one pluvial), both the ORV and LBDA underestimated wet and dry extremes. However, the ORV reconstruction better matched the distribution of soil moisture values and the spatial patterns of the instrumental data, particularly for the extreme values, compared to the LBDA reconstruction (Figure 4; Supplemental Figures 2–4). For droughts, the ORV consistently had lower mean absolute errors (differences ranging from 0.21 to 0.41) compared to the LBDA (Figure 4; Supplemental Figures 2–4). However, for the pluvial event, the two reconstructions had similar mean absolute errors (difference of 0.03) with the LBDA being slightly smaller (Supplemental Figure 4). When examining the correlation in PMDI (instrumental or reconstructed) between all grid points as a function of distance, the ORV better matched the instrumental PMDI with a steeper decline in correlation across distance compared to the LBDA (Figure 5). The LBDA showed the most spatial autocorrelation with a gradual decrease in correlation across distance, while the instrumental had the least spatial autocorrelation with a lower correlation between close grid-points and more variability (Figure 5). The ORV better matched the overall pattern and variability of the instrumental PMDI across distance but had more spatial autocorrelation (Figure 5).

In general, the probability distribution function (PDF) of the ORV reconstruction had a lower occurrence (densities of 0.17 compared to 0.23) of near-average years but higher densities (differences ranging from 0.01 to 0.05) for extremes, particularly drought, compared to the LBDA (Figure 6). The ORV distribution was nearly identical to the instrumental while the LBDA had lower densities of extremes (Figure 6). Similarly, the ORV had a larger number of reconstructed drought (median difference of 9 years)

conditions that better matched the instrumental record. The pluvial conditions were closer between the three datasets with the LBDA having the highest median and the instrumental the lowest median (Figure 6). Due to the larger number of extreme drought years, the ORV reconstructions had more frequent flips according to the flip index values compared to the LBDA (Figure 7). The central and southeastern portions of the region, in particular, showed a greater number of wet, dry, and total flips, resulting in ~30 more wet and dry flips and ~60 more total flips (Figure 7).

### 3.3 *Species Contributions*

With the highest average correlation values, *Quercus* spp. chronologies were consistently the strongest contributors to reconstruction models (Figure 8). The white oak (*Q. alba*) chronology from Lincoln's New Salem in Illinois had the highest JJA correlation value of 0.749, and as a species, *Q. alba* was the strongest species contributor (Figure 8). In addition to *Quercus* spp., black walnut (*Juglans nigra*) had an exceptionally high average correlation value, ranking the third highest. White ash (*Fraxinus nigra),* tuliptree (*Liriodendron tulipifera),* and sugar maple (*Acer saccharum*) were also strong contributors to drought models, with median correlation values greater than 0.38 (Figure 8).

### 3.2 *ORV and LBDA Validation Statistics*

Comparing how well each reconstruction model represented the instrumental data, we find that the variance explained ($R^2$-values) in the calibration and verification periods match well for the northern portion of the network, with values ranging from 40 to 60 percent variance explained (Figure 8). However, the ORV models for the southern half of the region generally explain less variance compared to the LBDA (Figure 9). Interestingly, the RE- and CE-values between the two reconstructions are generally more similar, with the ORV having poorer validation statistics in the southernmost portion of the region and the LBDA having weaker statistics in the central portion of the region (Figure 9).

Previous work has shown that radial growth from trees in the south-central portion of the region are becoming less sensitive to soil moisture compared to earlier time periods (Maxwell *et al.*, 2016). The comparison between a point-by-point reconstruction that ended in 1980 to a reconstruction that ended in 2010 demonstrates that while the calibration $R^2$-values are similar, the 2010 verification models explain

much less variance in the southern portion of the ORV (Figure 10). These are the same regions in the ORV reconstruction that explain less variance than the same gridded reconstructions of the LBDA. Importantly, the ORV 1980 and 2010 reconstructions used the same tree-ring chronologies (Figure 10). Therefore, our results indicate that tree rings in the southern portion of our study region have become less responsive to soil moisture.

## 4 Discussion

### 4.1 *ORV and LBDA Extreme Year Comparisons*

Tree rings have long been used to provide an historical context to hydroclimatic extremes (Stahle and Cleaveland 1994; Woodhouse and Overpeck, 1998; Cook *et al.*, 1999; Cook *et al.*, 2010; Pederson *et al.*, 2013). However, in some regions in the US, the tree-ring sites are sparsely distributed, and it is unknown what kind of impact that has on the representation of past climate. Due to the higher density of tree-ring chronologies and the smaller search radius (250 km for the ORV compared to 450+ km for LBDA) of the PC regression models when determining the pool of predictors, the ORV better replicates the spatial variability of the instrumental data compared the LBDA (Figure 4–5; Supplemental Figures 2–3). By using a ≥450 km radius for potential tree-ring chronologies, the LBDA was successful at reconstructing soil moisture even in areas that have a limited number of tree-ring chronologies. However, this approach results in the use of the same tree-ring chronologies in multiple grid points, spatially smoothing the variability of the reconstructed PMDI compared to the instrumental data (Figure 5). The same is true of the ORV; however, the increase in the spatial density of the chronologies allows a smaller search radius and therefore, can increase the spatial variability in the ORV (Figure 5). The increase in spatial variability in PMDI values of the ORV better matches the instrumental data while still providing a statistically valid reconstruction model (Figure 4–5; Supplemental Figures 2–4). These findings have important implications, particularly in regions with a sparse tree-ring network where the LBDA or other drought atlases likely underestimate localized droughts and pluvials. Increasing the spatial density of the tree-ring network will allow a more accurate spatial representation of extreme events nearly anywhere where trees are sensitive to climate.

In addition to the increase in spatial variability of extremes that we find, previous work suggests increasing the density of the tree-ring network can uncover previously unknown droughts and pluvials at more local scales (Maxwell and Harley, 2017; Strange *et al.,* 2019; Pearl *et al.* 2020). Here, we find the support of better-localized representations of extremes by increasing the density of the tree-ring network with the ORV having a larger number of droughts and pluvials compared to the LBDA (Figure 6). The increase in extremes has important implications on the long-term variability of past hydroclimate and to the interannual volatility of PMDI. Recent work has shown increases in interannual volatility has important impacts on agriculture (Locke *et al.*, 2017), and social and ecological systems (Casson *et al.*, 2019). Our finding suggests that in areas with a sparse tree-ring network, such as in the ORV, tree-ring reconstructions underestimate extremes and therefore, volatility in extremes is also underestimated. By increasing the density of the network and better representing localized extremes, we find a higher number of flips (Figure 7). The better representation of localized extremes results in a more accurate representation of past climatic volatility and can be used to better place current and future projected changes into context. With gridded reconstructions of both soil moisture and temperature becoming more common with the increase in available tree-chronologies (*e.g.,* Anchukaitis *et al.,* 2017; Morales *et al.,* 2020; Pearl *et al.* 2020), we show the importance of valuing higher density from a larger range of species within the network in addition to the length of the chronologies.

### 4.2 *Species Contributions*

Historically, soil moisture reconstructions from tree rings in the eastern US have been dominated by a few species, such as *Q. alba,* baldcypress (*Taxodium distichum*)*,* eastern hemlock (*Tsuga canadensis*) (Zhao *et al.,* 2019). In addition to increasing the spatial density of the network, the ORV reconstruction has increased the number of species used, many of which are co-occurring. The use of multiple species has been shown to increase model performance (Pederson *et al.*, 2001; Frank and Esper, 2005; Cook and Pederson, 2011; Maxwell *et al.*, 2011; Pederson *et al.*, 2012, Maxwell *et al.* 2015). Examining the correlation values of the species used in the reconstructions models, *Quercus* (oak) species in general, contribute more to the models (Figure 8), which is part of the reason why they have been traditionally used so frequently. However, we find that several species, including *J. nigra, L. tulipifera*, and *A.*

*saccharum* among others, make strong contributions to the model as well (Figure 8), further supporting that these species are sensitive to hydroclimate variability (LeBlanc *et al.* 2020; Au *et al.,* 2020). These findings agree with recent studies that suggest less commonly used species can increase the representativeness of tree-ring reconstructions of climate (Pederson *et al.,* 2012; Maxwell, 2016; Maxwell and Harley, 2017; Alexander *et al.,* 2019).

### 4.3 *ORV and LBDA Validation Statistics*

While increasing the spatial density of the tree-ring network allowed the reconstructions to more accurately capture the spatial variability of extreme conditions, the reconstruction models of the ORV have less predictive skill compared to those of the LBDA, especially during the verification period (Figure 9). The two networks have some overlap in chronologies, but while the ORV has a higher density of chronologies within the Ohio River Valley region, the LBDA can draw from more chronologies across a larger region. While the larger radius increases the number of samples in the model and could lead to more explained variance for the LBDA, the ORV reconstruction better spatially replicates extremes in the instrumental period (Figures 4; Supplemental Figures 2–4).

Interestingly, the decrease in variance explained in the southern portion of the region may not attribute from differences of sample depth in the tree-ring network. When using the same chronologies while ending the calibration period at 1980 instead of 2010 for the ORV reconstruction, the validation statistics compare very well with the LBDA. However, by updating the chronologies to 2010, the $R^2$ and the validation statistics drop dramatically for the grid reconstructions in the southern portion of the region (Figure 10). These findings support Maxwell *et al.* (2016), where they found trees in this region to have a weakening signal to soil moisture, termed the "Fading Drought Signal." The recent decrease in sensitivity of tree growth to soil moisture has also been documented outside of the ORV, in the Mid-Atlantic US (Helcoski *et al.,*2019), indicating the impact of a changing climate could influence the representation of tree rings to climate in mid-latitude locations. Drought in the Midwest during the instrumental period (1901–2010) was temporally clustered in the 1930s and 1950s. The only recent droughts in the study period were in 1988 and 2002. In both cases, the northern portions of the region

experienced severe drought (in excess of -4.0 PMDI values for 1988), but the southern portion of the region only experienced moderate dryness (PMDI values of ~ -2.0). Maxwell *et al.* (2016) attributed the weakening signal to a recent period without severe drought; however, Helcoski *et al.* (2019) discussed the possibility of increases in carbon dioxide concentrations in addition to a long period of wetness interacting to weaken tree growth responses to soil moisture. However, recent works examining the simultaneous influence of water availability, carbon dioxide concentrations, and acidic deposition found that water availability was the leading influence on tree growth (Levesque *et al.*, 2017; Maxwell *et al.*, 2019), suggesting a wet period is likely driving the weakening signal. The decreasing performance of the southern reconstructions support these findings as this region has been generally wet and absent of severe drought. While Maxwell *et al.* (2019) found that *Acer* species had a more stable relationship with soil moisture, and *A. saccharum* was a strong performing species in the reconstructions models, the inclusion of multiple, co-occurring *A. saccharum* records did not dramatically influence the validation statistics of the reconstruction models in the southern portion of the region. Our findings demonstrate the complexity of tree species responses to rapidly changing climate regimes and stress the need to better understand species responses to changing climate and determine what impact those responses could have on reconstructions of soil moisture.

**5 *Conclusions***

By increasing the density of the tree-ring network in a region that is poorly represented in the LBDA, we created a gridded PMDI reconstruction for the ORV region. We compared our gridded reconstruction with the LBDA and found that increasing the density of the tree-ring network resulted in an increase in localized hydroclimatic extremes that better match the spatial and temporal patterns of the instrumental data. However, calibrating our models with more recent data (up to the year 2010) resulted in a decrease in variance explained and validation statistics for the southern portion of the region. This region has not experienced extreme droughts recently, which is likely driving the decrease in model performance. Increasing spatial density of the tree-ring network is important to better represent localized extremes in the past, indicating that researchers should continue to target previously unsampled old-growth forests. Similarly, the time in which the trees are sampled is also important to model performance. Long periods

without extreme hydroclimate variability can result in reconstruction models that are less representative of climatic conditions. We stress the need to update previously-sampled chronologies to the current period so that longer calibration models can have the chance to better represent the range of sensitivity of trees rings to climate. Further, more work is needed to extend more of the ORV chronologies to better represent climate further in the past. Targeting wood from historical structures and combining with surrounding living chronologies of the same species could be one way of achieving longer chronologies in this region (Harley et al. 2011; Matheus *et al.* 2017). Overall, we find that a higher spatial density of the tree-ring network will improve the local representation of reconstructed climate. However, more work is needed to better quantify how the strength of the relationship between tree growth and climate varies through time.

Data Availability: All reconstructions will be uploaded onto the NOAA paleoclimate page. All tree-ring chronologies used in this manuscript will be uploaded to the International Tree-Ring Databank.

Author Contributions: JTM and GLH designed the methods of the manuscript. JTM performed analyses with feedback from GLH. TJM, BMS, KVK, and TFA helped develop tree-ring chronologies with assistance from JTM and GLH. All authors contributed to data collection and the preparation of the manuscript.

Acknowledgments: The collection of tree-ring samples was partially funded by a USDA Agriculture and Food Research Initiative grant 2017-67013-26191 and from the Indiana University Vice Provost for Research Faculty Research Program. We would like to thank James Dickens, James McGee, Josh Oliver, Karly Schmidt-Simard, Brynn Taylor, Michael Thornton, Senna Robeson, Matt Wenzel, and Luke Wylie for assistance in the field and laboratory.

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

Figures:

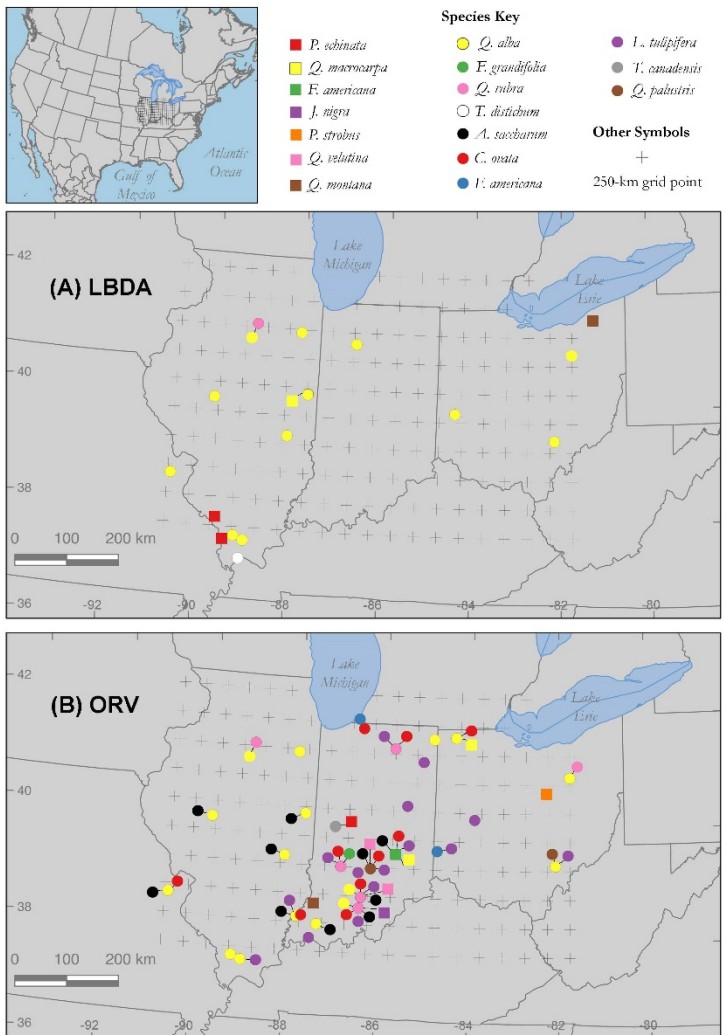

Figure 1: Map of 0.5° x 0.5° PMDI grid points (*n* = 181) across the Ohio River Valley (ORV) region, Midwest US—defined as 37.75–42.25° N, 90.75–82.25° W—plotted with tree-ring chronology sites included from the (A) ITRDB and (B) ORV networks. Sites with single-species and multiple-species are denoted by symbol shape and color (see Supplemental Table 1). Note: most ITRDB sites consist of single species in the LBDA but multiple species are represented in the ORV.

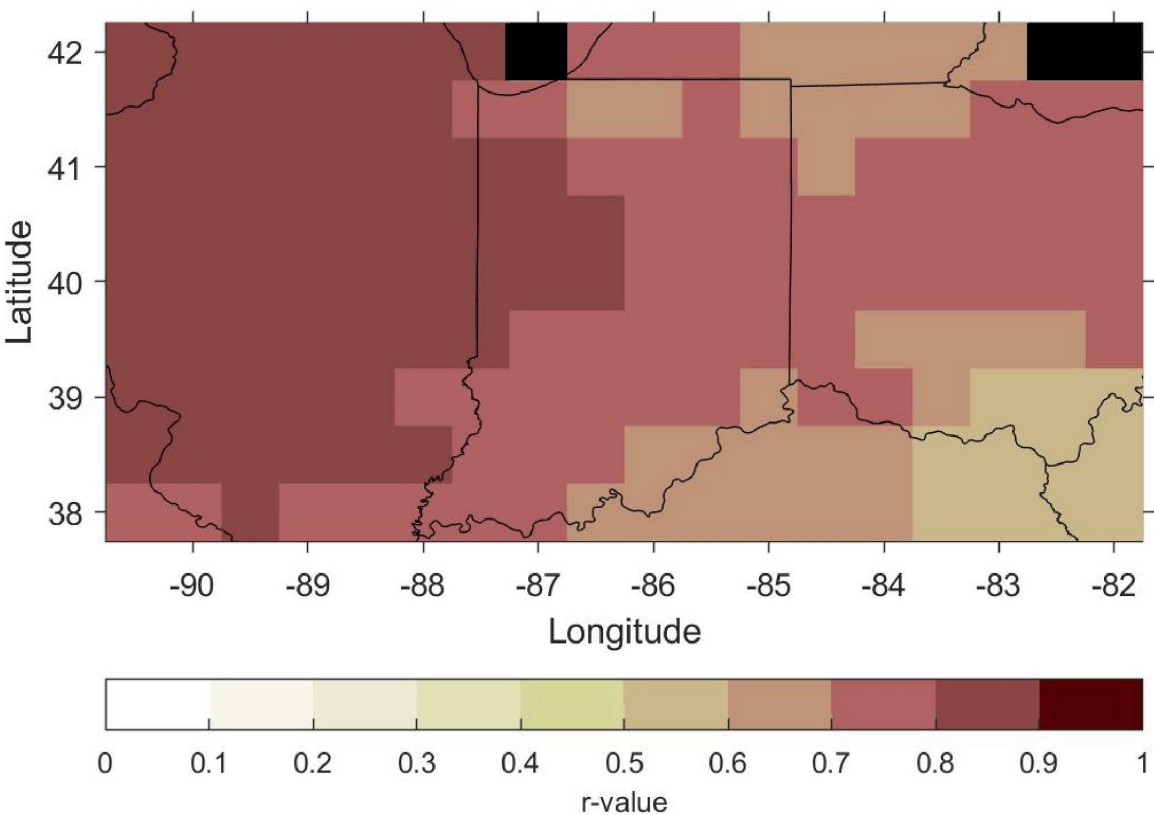

Figure 2: Map of correlation values between the LBDA and ORV reconstruction during the period of
1830–2005. The correlations of each grid shown in the map are all significant at the 0.05-level. The black
cells represent locations over the Great Lakes and therefore, no data is available for correlation analysis.

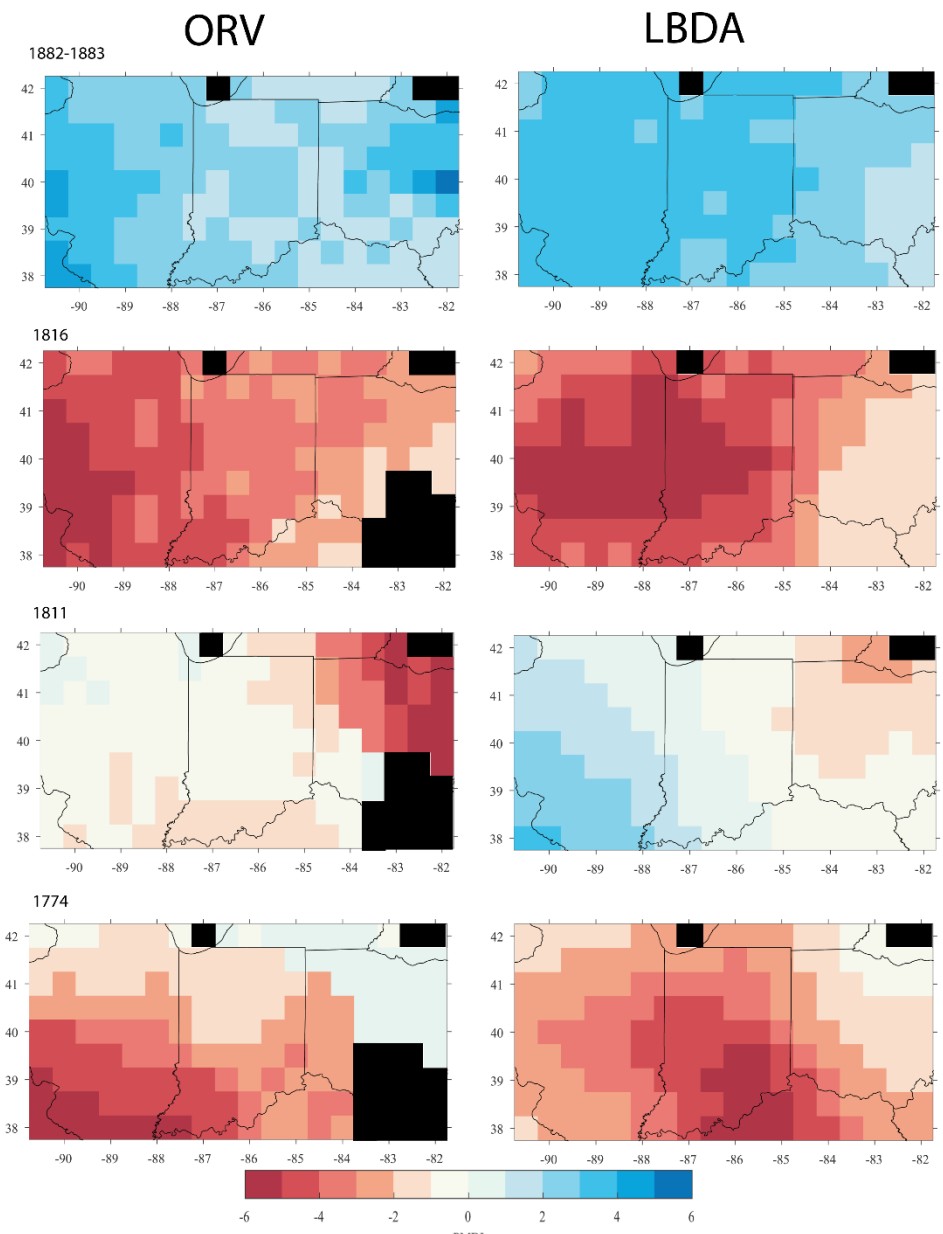

Figure 3: Spatial comparison of the ORV (left column) and the LBDA (right column) of reconstructed PMDI during years that experienced hydroclimatic extremes. Red cells represent below average PMDI and blue cells represent above-average PMDI. Black cells represent no data either due to being over water or from not having at least five chronologies to create a reconstruction.

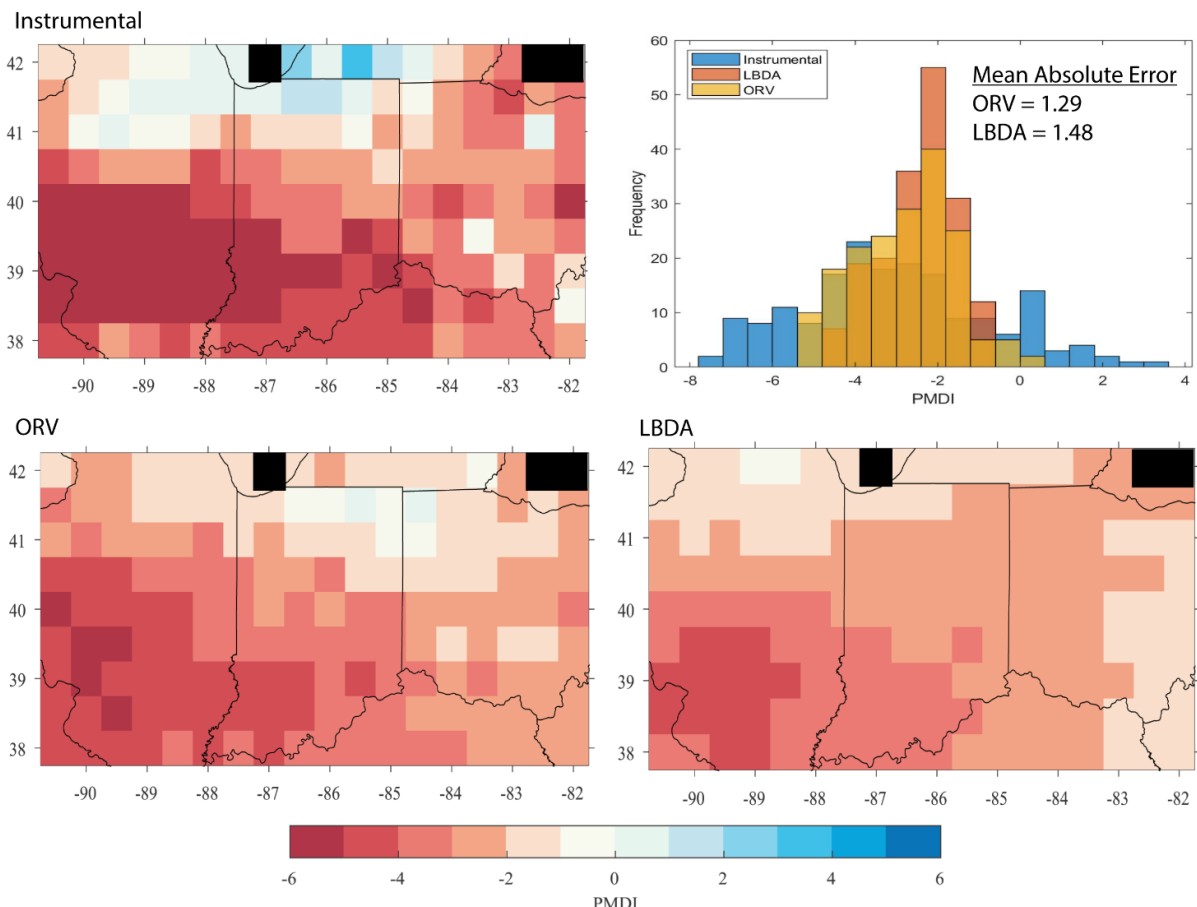

Figure 4: Maps showing PMDI values for the instrumental data, ORV, and LBDA reconstructions for the year 1954. The histogram represents the frequency of PMDI values for the instrumental, ORV, and LBDA PMDI values. The mean absolute error values show that the ORV reconstruction more accurately matches the instrumental data compared to the LBDA reconstruction. Black grids represent areas over water and therefore, no data.

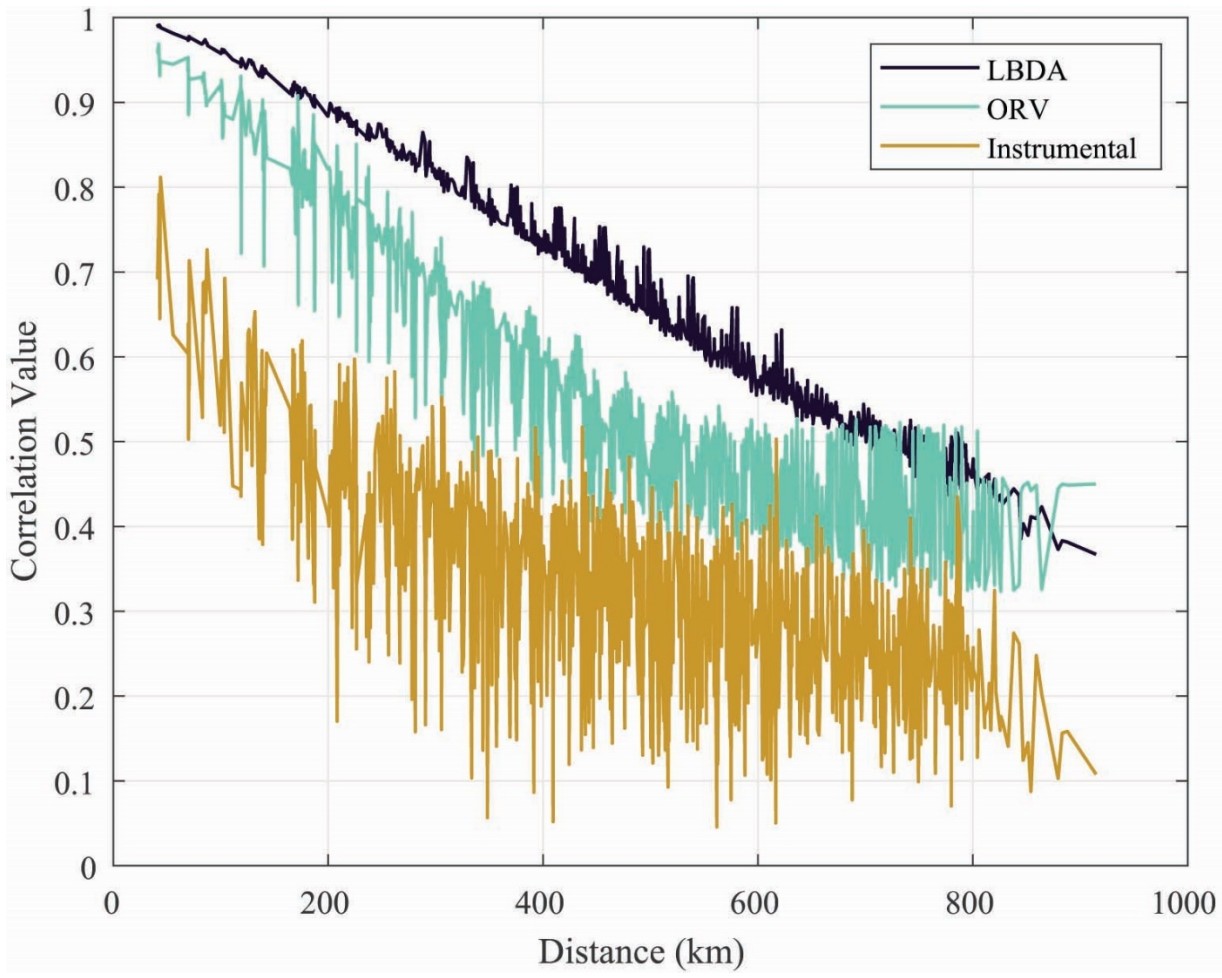

Figure 5: Average correlation coefficients between PMDI values across all grid-points as a function of

distance. LBDA and ORV are reconstructed PMDI values.

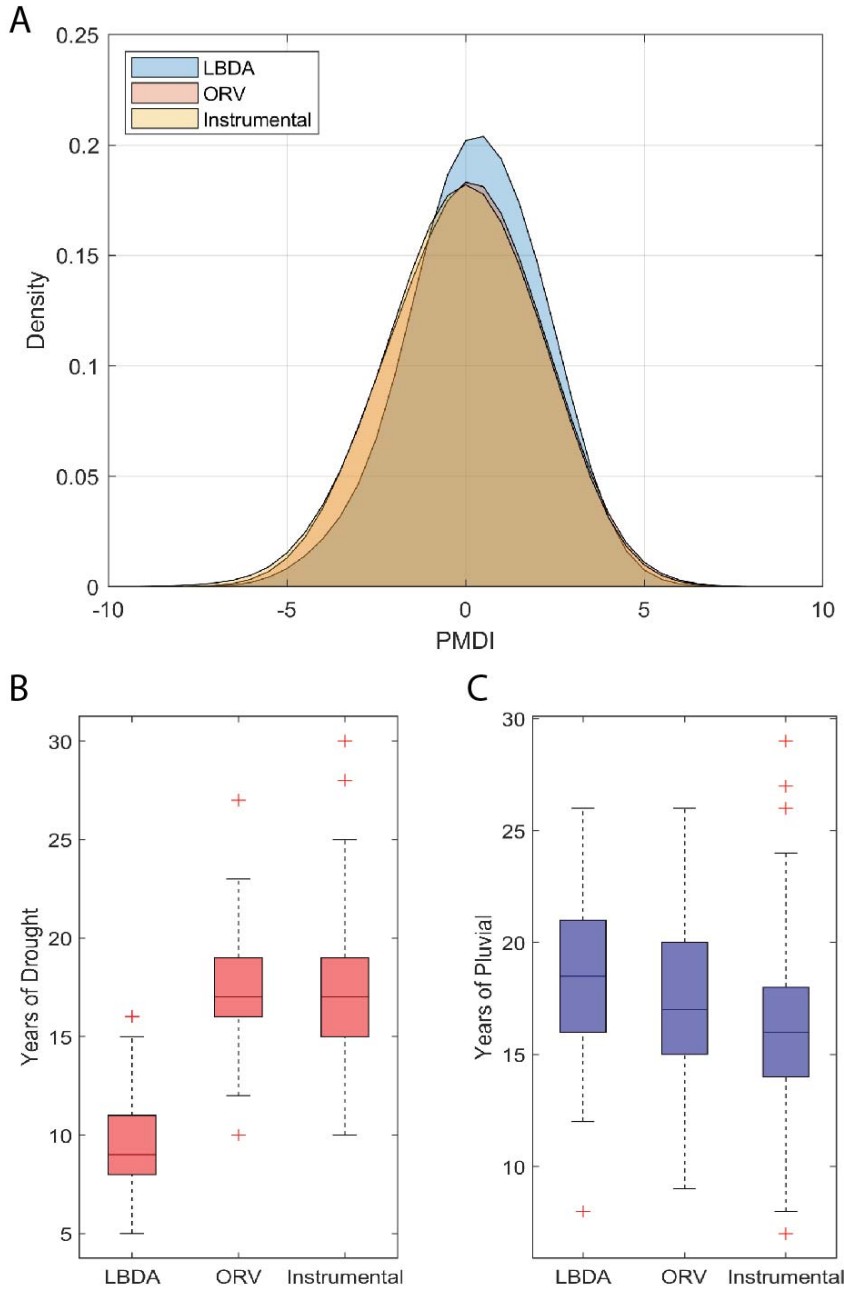

655

Figure 6: A) Probability distribution functions for all gridded reconstructed PMDI values for the ORV and LBDA networks. B) Boxplot of the number of droughts (PMDI ≤ -2.0) years between LBDA and ORV. C) Boxplot of the number of pluvials (PMDI ≥ 2.0) years between LBDA and ORV.

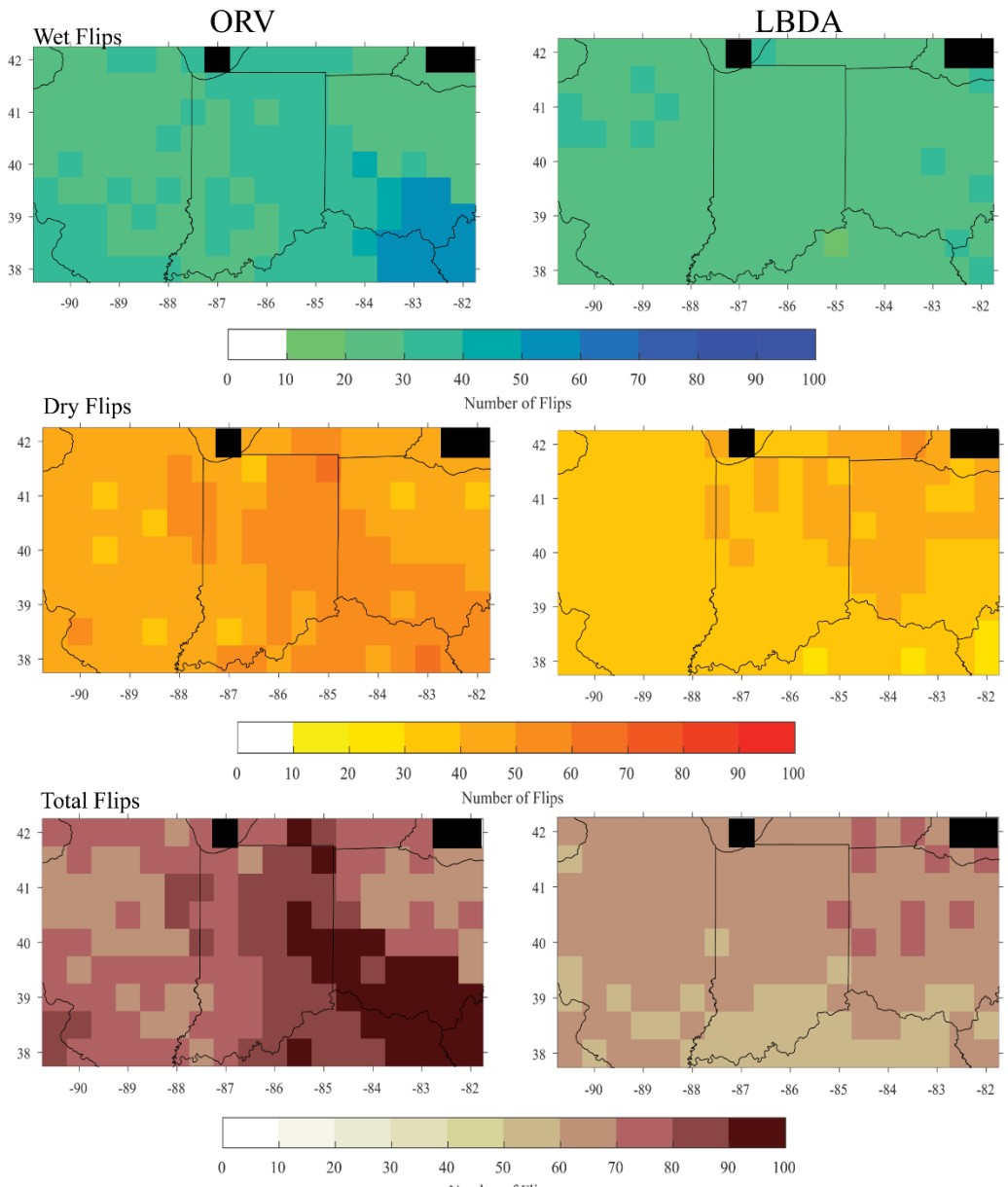

Figure 7: Maps of the number of wet flips (top row), dry flips (middle row), and total flips (bottom row), for the ORV (left column) and the LBDA (right column). Black cells represent values over water and therefore, no data.

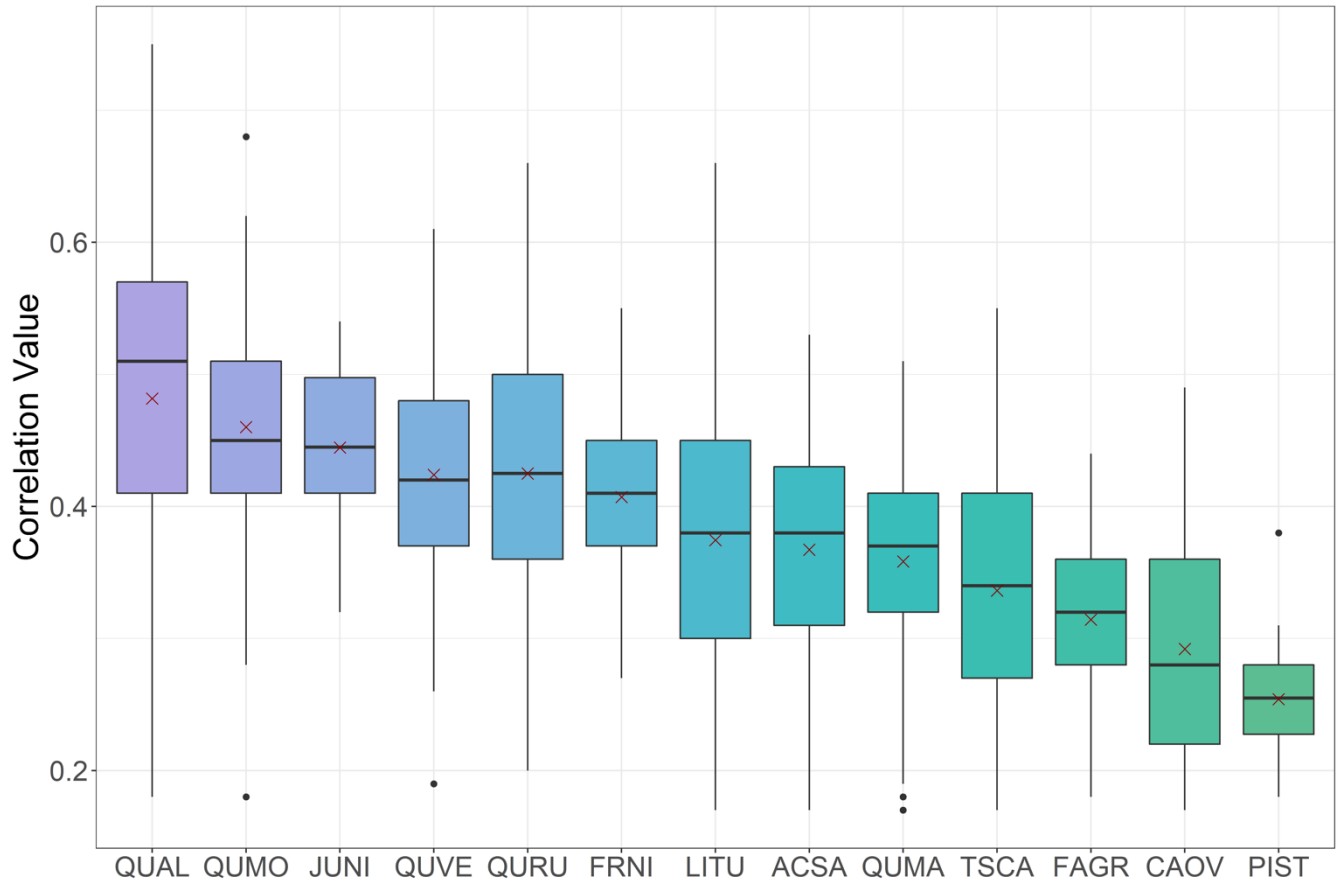

Figure 8: Correlation values between species chronologies and PMDI for the gridded reconstruction models. The "x" represents the mean beta weight for the species. QUAL=*Q. alba,* QUMO=*Quercus montana,* JUNI=*Juglans nigra,* QUVE=*Q. velutina,* QURU=*Q. rubra,* FRNI=*Fraxinus nigra,* LITU=*Liriodendron tulipifera,* ACSA=*Acer saccharum,* QUMA=*Q. macrocarpa,* TSCA=*Tsuga canadensis*, FAGR=*Fagus grandifolia*, CAOV=*Carya ovata*, and PIST=*Pinus strobus.* The species are ranked by their mean correlation values from highest to lowest.

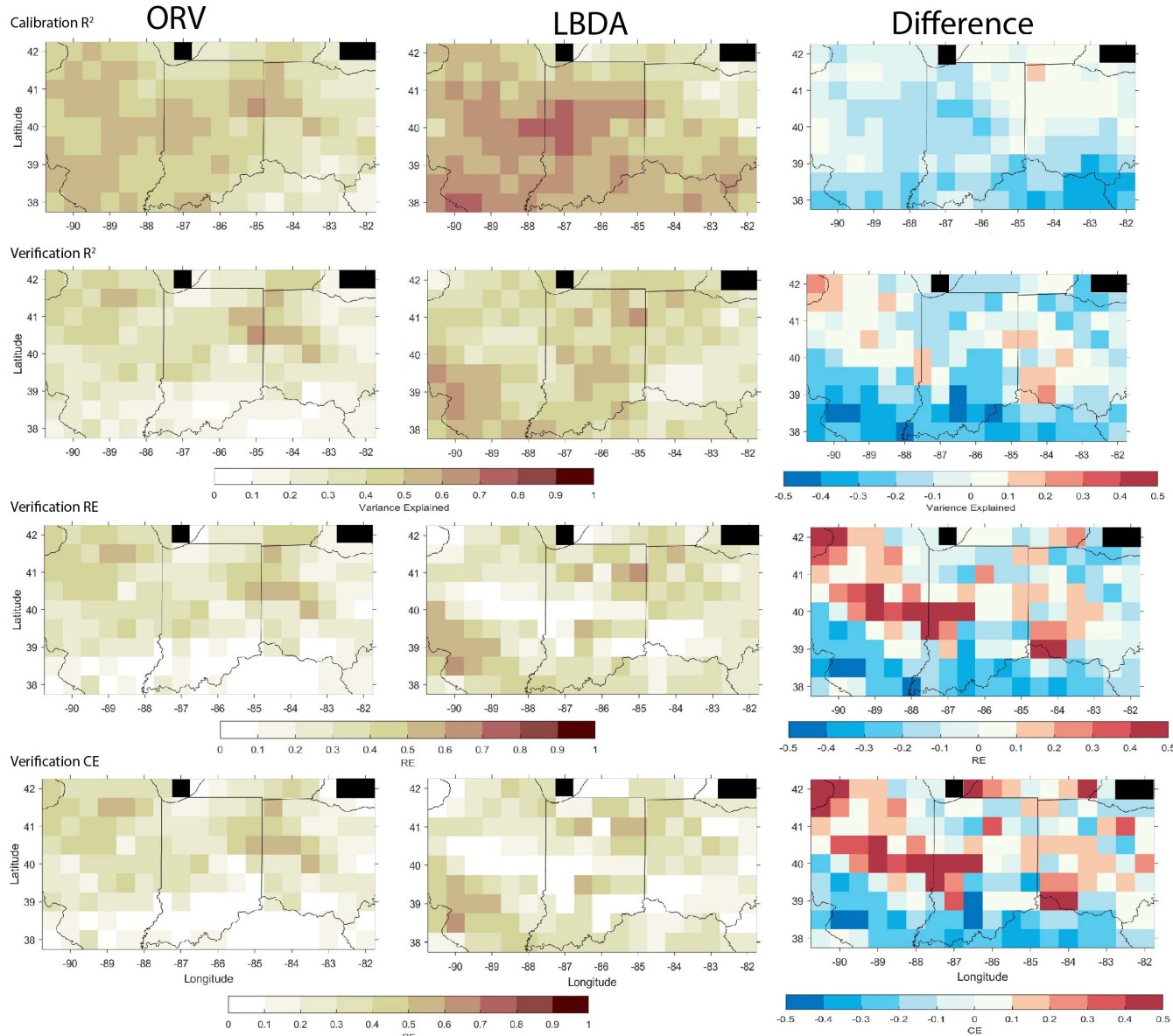

Figure 9: Comparison of the calibration (1901–1955) and validation (1956–2010) statistics between the ORV (left column) and LBDA (right column) reconstructions. Difference represents LBDA values subtracted from ORV. Black cells represent values over water and therefore, no data.

s

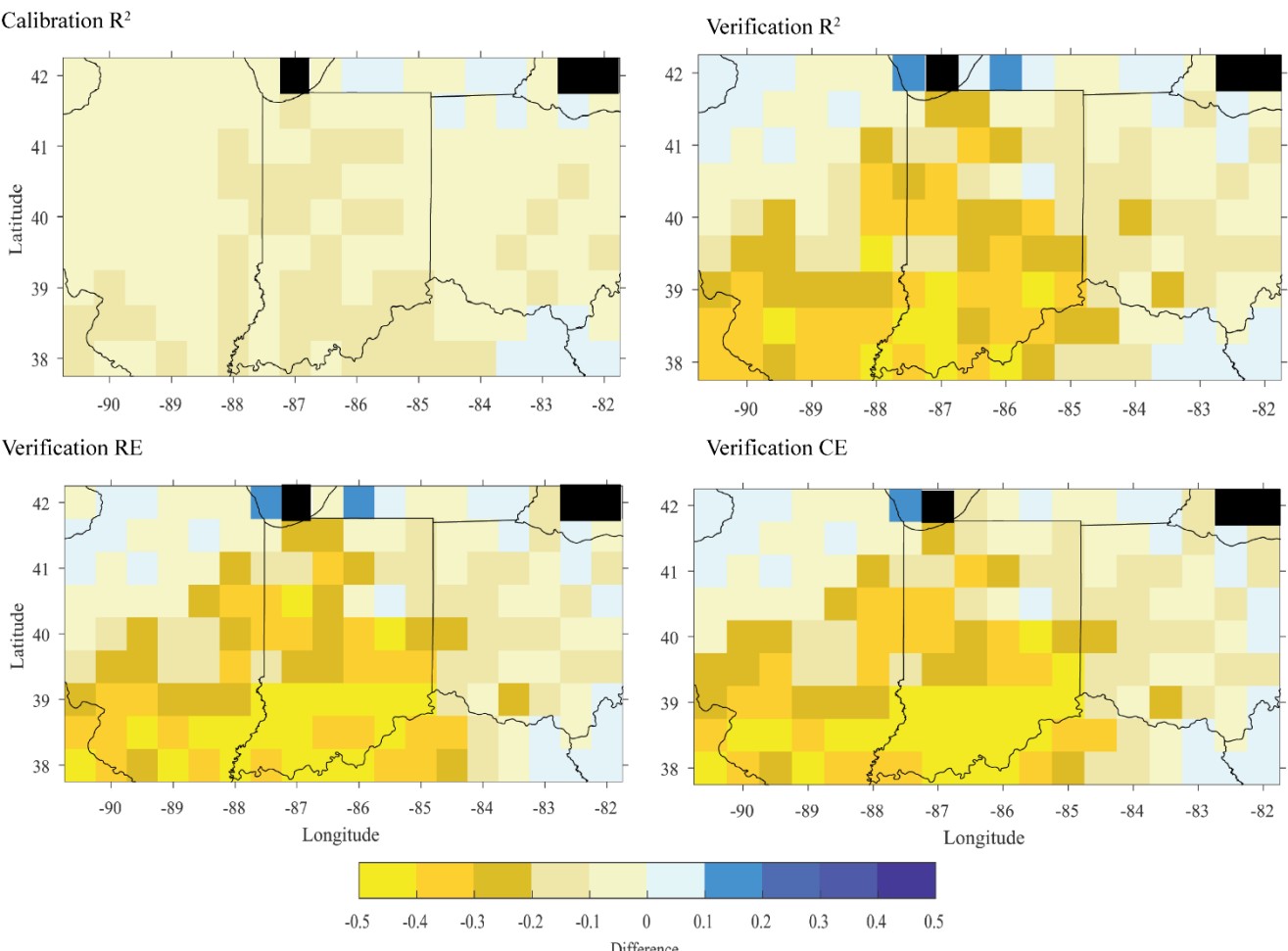

680 Figure 10: Maps of the difference between the ORV reconstruction when ending the calibration period in 2010 compared to 1980 (*i.e.,* $ORV_{2010} - ORV_{1980}$) for calibration $R^2$, verification $R^2$, RE, and CE. Black cells represent values over water and therefore, no data.