# Peer review of "Sampling density and date along with species selection influence spatial representation of tree-ring reconstructions"

_Climate of the Past, 2020_

## Referee Comment (RC1) · Anonymous Referee #1 · 6 Apr 2020

Review for Maxwell et al 2020

"Sampling density and date influence spatial representation of tree ring reconstructions" uses an updated, multi-species, tree-ring network in the Ohio River Valley to demonstrate the influence of increased predictor density and record length on spatial drought reconstructions. This paper presents well-supported findings that increasing predictor density in a gridded hydroclimate reconstruction can identify more localized patterns and emphasizes that the hydrologic sensitivity of some of the species is changing. The authors discuss the influence of recent dampening of extreme droughts and pluvials compared to the 1900-1980 time period, and how incorporating non-traditional dendroclimatology species can strengthen the reconstructions.

The clearly laid out discussion showed the power and limitations or increasing the

predictor density in a spatial reconstruction. The authors' conclusions regarding the "fading drought signal" of trees is important for future hydroclimate reconstructions – particularly in this region.

This manuscript should be published, with a few minor edits. A multi species approach to climate reconstructions, comparing the NADA to a denser predictor network in data-scare areas (most recently in Pearl et al., 2019), and adjusting the calibration/verification time period for reconstructions have all been done previously in other regions of the U.S. . Thus, the main novelty of this study is the site geography. The discussion, therefore, would benefit from the inclusion of the author's thoughts on a climatological explanation for the higher number of flips at a local scale compared to large scale. That is, why does the Ohio River Valley experience these flips, how is this distinct from the large scale regional dynamics that "smooth" out these flips in the lower resolution reconstruction?

Minor comments:

Be careful when hyphenating "tree ring". It should be "tree-ring" when used as an adjective or modifier, and "tree ring" when use as a noun or direct object. E.g. "tree-ring reconstruction"

Title: The title is a bit convoluted for me. . . the paper also looks at species not just sample density and length of the record. Perhaps "Sampling characteristics" or "Predictor characteristics influence climate patterns/phenomena in tree-ring reconstructions:

Line 45: add "of climate" after reconstructions so that readers know you are not reconstructing the mechanisms mentioned at the beginnings of the sentence.

Line 47: delete "historical" – tree rings provide context in the prehistory too.

Line 48: "instrumentally -recorded" is awkward. "Droughts and pluvials captured in the instrumental record. . . "or similar

Line 77: Pearl et al., 2019 did this in New England

Line 274: Again, I think either "historical" or "past" should be here, both are redundant

Line 295: I would also cite Pearl et al, 2019

Line 319: Alexander et al 2019 also saw this in a temperature reconstruction

Line 275: replace "but" with "be"

Figure 1: Suggestion to move the USA map and species symbols to the top of the figure. Its odd that its in between panel "A" and "B"

Figure 3: I would choose either contour lines or un-smoothed squares. Since the maps are not "filled" or "smoothed" the contours are unnecessary and distracting. Reference the color bar in the caption. I also suggest having white (not green or yellow) as 0 PMDI, and then hatch out the grid cells with no data.

Figure 4: same comment as 3

Figure 6: suggestion to have all color bars go from white to color and then hatch out the insignificant/no data values. Solid blocks of color are more difficult with color blindness.

Figure 8: Mention what calibration time period is represented in this figure.

Figure 9: Again, suggestion to NOT have green has the zero value, have white.

Figures in general, increase the size of the text

Suggested citations:

Alexander, M. R.,J.K. Pearl, D. A. Bishop, E.R Cook, K.J. Anchukaitis, N. Pederson, The potential to strengthen temperature reconstructions in ecoregions with limited tree line using a multi species approach, Quaternary Research, 1-15, doi: 10.1017/qua.2019.33, 2019

Pearl, J.K., K.J. Anchukaitis, N. Pederson, J. Donnelly, Multivariate climate field reconstructions using tree-rings for the northeastern United States, Journal of Geophysical Research – Atmospheres, doi:10.1029/2019JD031619, 2019

---

## Referee Comment (RC2) · Anonymous Referee #2 · 4 Jun 2020

The work by Maxwell et al. explores whether an improvement of the tree-ring based Living Blended Drought Atlas (Cook et al., 2010) can be achieved over the Ohio River Valley, US, by increasing the density of the proxy network, as well as incorporating a broader range of tree species. The work also briefly assess whether including the last decades in the calibration/validation exercise might change the performance of the reconstruction models.

Overall I find the ideas of this work compelling, and thus I regard the results being of general and international interest. The data is treated with more or less standard methods within the field of dendrochronology and the analyses appear to be sound. Moreover, I find the idea of combining multiple species to obtain a more robust reconstruction compelling. Although the study has a sound rationale and execution, the

value gained by the authors' approach appear to have had marginal benefits. In light of this, I miss a discussion on quality versus quantity of predictors in state-of-the-art field reconstructions. Listed below, are a few more specific comments which I hope will help the authors improve the final manuscript.

Specific comments:

P1/L32: "By sampling tree in 2010 [. . .] " reword to "By extending the calibration period to 2010 [. . .]" (suggestion)

P2/L50: Oliver et al., 2019 is missing in the reference list. Also, tree-ring based drought reconstruction are not restricted to the mid-latitudes and certainly not only to US (which is somehow implied by the references the authors cite)

P2/L54: "[. . .] creating a 2.5° x 2.5° reconstruction" was it not a 2° x 3° PDSI grid that was used in Cook et al., 1999?

P2/L55: "The NADA produced multiple centuries of both spatial and temporal data of drought variability" remove either "multiple centuries" or "temporal" – its redundant to have both

P3/L81: "Yet, developing a reconstruction assumes that this climate-tree-growth relationship is stationary over time. This assumption was generally true in the early development of the field of dendrochronology (ca. 1920s–1950s; Fritts, 1976). However, as human activities drive the Earth's climate system into historically unprecedented, and potentially non-stationary and non-analogous conditions (Milly et al., 2008), exceptions to this assumption have emerged." Please rephrase, it's unclear and contradictory. By pointing out that the system is stationary between 1920s-1950s the authors also admit that the relationship is non-stationary.

P3/L86: "Changes in the drought signal recorded by tree rings have been established only recently [. . .]" Do the authors refer to the midwestern US? If so, please be more specific.

P4/L97: "... if the year when trees are sampled influences the climate reconstruction" This sentence is awkward, please rephrase. It is not the year when the trees are sampled, but rather the period that is covered by the calibration period that might influences the reconstruction skill.

P4/L98: "We calibrate the reconstruction with recent (post-1980) radial growth and climate data ..." Please rephrase. It is not the reconstruction that is calibrated, but the tree-ring data that is calibrated with climate data to obtain the reconstruction.

P5/L126 "We used the list method to visually crossdate all samples, and then the program COFECHA to statistically verify the crossdating" (suggestion)

P5/L147: indicate the period for the correlation analysis, and also the significance level applied in this screening

P6/149: what is the rational behind using a 250-km search radius? Was it selected based on the spatial characteristics of regional drought climatology observed in the instrumental data?

P6/L150: it should be mentioned that a dynamic search radius was used to produce LDBA, with the requirement that at least five chronologies had to be located around each grid point. By eyeballing the very sparse tree-ring network in fig 1A I would assume that the search radius might even had been larger than 450 km across the ORV region. Please check if this is the case, and clarify in the text.

P6/L151: did the authors also consider lagged associations between tree-ring and drought data? This has been done in LDBA (i.e. tree-ring data year t + 1 were considered in the reconstruction of drought in year t), meaning that there were actually potentially twice as many predictors of drought at each grid point compared to the number of tree-ring chronologies located around each grid. Also, were there any requirements about the minimum no of chronologies to be included in the predictor pool for the new ORV reconstruction?

P6/L161 and P9/L239, L295: The spectral properties of the resulting hydroclimate reconstructions will be affected by the way the short-lag autocorrelation structure in the tree-ring data is treated. If I am not mistaken, for the LDBA reconstruction a low-order AR model was fitted to both the instrumental and tree-ring series to correct for the mismatch in the short-term autocorrelation. The prewhitened time series were then used to test the association between drought and tree-growth and to build the regression models. The autocorrelation of instrumental data was then added back to the final tree-ring reconstructions of drought. It should be mentioned if a similar approach was adopted also in this study.

P8/L220: "The ORV reconstructions were shorter in length (maximum of 343 years) compared to the LBDA reconstructions (maximum of 2,006 years) due to each grid reconstruction having a smaller search radius (250 km vs 450 km) for chronology in-clusion." This sentence needs to be rephrased. The ORV reconstruction is not shorter because of a smaller search radius, but because the temporal extension of the tree-ring network was more limited than in the LDBA.

P9/L248: Not sure I understand how the beta-weight values for the different species were obtained. Are these the loadings from the PCA?

P11/L280 "compared to"

P11/L283: "multiple gridded reconstructions" perhaps "multiple grid points" would be better suited here

P14/361: "[...] calibrating our models " do the authors mean validating?

Conclusions: I am missing a sentence or two about future prospects/possibilities of extending the newly sampled data in the ORV region back in time.

Figure 1: please add a scale ruler for reference. Also, it might not be clear what the rectangle in the figure represents.

Figure 4: the spatial patterns in the ORV and LBDA reconstructions look pretty similar

to me. I would therefore be careful to conclude, based only on this plot (as well as figs 1-3 in the supplement), that the ORV reconstruction better match the distribution of soil moisture values and the spatial patterns of the instrumental data compared to the LBDA reconstruction" (L233). The authors need to perform some additional analysis to support this conclusion. For instance, the authors could compute point-by-point correlations between all possible pair of grid points in the instrumental data, ORV and LBDA, respectively, and then plot the correlation as a function of distance between grid points (correlation decay distance). If the spatial characteristics of droughts in the ORV reconstruction is indeed more accurate than in the LBDA, then the CDD of the ORV would be more similar to instrumental data. The slope of the correlation vs distance curve would be much less steep for the LBDA reconstruction, because of higher spatial autocorrelation

Figure 5: the information in this figure loses some of its value if not compared /validated against the spectral properties of the instrumental data. This could be done by restricting the analysis to the modern period when also instrumental data is available

Figure 6: please indicate in the different figures whether the flips refers to wet, dry or total flips.

Figure 8: add the periods for calibration and verification either in the figure or in the caption. Also, the figures is not easily interpreted. I suggest the authors add a third column where the differences/residuals between ORV and LBDA calibration and verification statistics are shown.

P7/L200 mentions that the 1941-1980 period was used for validation, while in fig 9 caption it says that calibration period ended 2010. Please clarify.

Not all the text is visible in the supplemental Table 1. The timespan of the chronologies should be included. Also, the state abbreviations would probably be meaningless for most of the international readership (at least they should be defined in the caption if the authors decide to keep them)

There are two figure 3 in the supplement

---

## Author Comment (AC1) · 9 Jun 2020

We would like to thank the Referee for the constructive review. The feedback provided will help us to improve the manuscript. Written below are our point by point responses to the Referee's comments. Our responses are below each comment and are the changes we propose to the manuscript based on the Referee's comments. The revised version of the manuscript will be prepared based on the decision of the Editor.

Anonymous Referee #1

COMMENT: "Sampling density and date influence spatial representation of tree ring reconstructions" uses an updated, multi-species, tree-ring network in the Ohio River Valley to demonstrate the influence of increased predictor density and record length

on spatial drought reconstructions. This paper presents well-supported findings that increasing predictor density in a gridded hydroclimate reconstruction can identify more localized patterns and emphasizes that the hydrologic sensitivity of some of the species is changing. The authors discuss the influence of recent dampening of extreme droughts and pluvials compared to the 1900-1980 time period, and how incorporating non- traditional dendroclimatology species can strengthen the reconstructions. The clearly laid out discussion showed the power and limitations or increasing the predictor density in a spatial reconstruction. The authors' conclusions regarding the "fading drought signal" of trees is important for future hydroclimate reconstructions – particularly in this region. This manuscript should be published, with a few minor edits. A multi species approach to climate reconstructions, comparing the NADA to a denser predictor network in data-scare areas (most recently in Pearl et al., 2019), and adjusting the calibration/verification time period for reconstructions have all been done previously in other regions of the U.S. . Thus, the main novelty of this study is the site geography. The discussion, therefore, would benefit from the inclusion of the author's thoughts on a climatological explanation for the higher number of flips at a local scale compared to large scale. That is, why does the Ohio River Valley experience these flips, how is this distinct from the large scale regional dynamics that "smooth" out these flips in the lower resolution reconstruction?

RESPONSE: Thank you for the positive comments on the manuscript. In the revision, we will include work from Pearl et al., 2019 and agree that expanding the discussion on the number of flips. The ORV reconstruction has a higher number due to better capturing the local extremes compared to the LBDA. We are picking up small but extreme droughts and pluvials as well as higher local extremes in broad scale drought and pluvial conditions. In the resubmission, we will look into the large-scale regional dynamics and see if we are picking up anything new compared to the LBDA but the likely differences are localized extremes.

Minor comments: COMMENT: Be careful when hyphenating "tree ring". It should be

"tree-ring" when used as an adjective or modifier, and "tree ring" when use as a noun or direct object. E.g. "tree- ring reconstruction"

RESPONSE: Thank you, we will check all uses and ensure we only use a hyphen when grammatically correct.

COMMENT:. . the paper also looks at species not just sample density and length of the record. Perhaps "Sampling characteristics" or "Predictor characteristics influence climate patterns/phenomena in tree-ring reconstructions:

RESPONSE: Great suggestion, we will change the title accordingly.

COMMENT: Line 45: add "of climate" after reconstructions so that readers know you are not reconstructing the mechanisms mentioned at the beginnings of the sentence.

RESPONSE: We will make this change

COMMENT: Line 47: delete "historical" – tree rings provide context in the prehistory too.

RESPONSE: We will make this change

COMMENT: Line 48: "instrumentally recorded" is awkward. "Droughts and pluvials captured in the instrumental record. . . "or similar

RESPONSE: Good suggestion, we will make this change.

COMMENT: Line 77: Pearl et al., 2019 did this in New England

RESPONSE: We will add Pearl et al. 2019 in the paper and add discussion where needed.

COMMENT: Line 274: Again, I think either "historical" or "past" should be here, both are redundant

RESPONSE: We will make this change.

COMMENT: Line 295: I would also cite Pearl et al, 2019

RESPONSE: We will make this change.

COMMENT: Line 319: Alexander et al 2019 also saw this in a temperature reconstruction

RESPONSE: We will add this citation and discuss where needed.

COMMENT: Line 275: replace "but" with "be"

RESPONSE: We will make this change.

COMMENT: Figure 1: Suggestion to move the USA map and species symbols to the top of the figure. Its odd that its in between panel "A" and "B"

RESPONSE: We will make this change.

COMMENT: Figure 3: I would choose either contour lines or un-smoothed squares. Since the maps are not "filled" or "smoothed" the contours are unnecessary and distracting. Reference the color bar in the caption. I also suggest having white (not green or yellow) as 0 PMDI, and then hatch out the grid cells with no data.

RESPONSE: We will change white color to represent 0 PMDI and then hatch the no data cells as suggested. We will explore a better way to illustrate the map based on the comment. As of now, we think we will just remove the contours.

COMMENT: Figure 4: same comment as 3

RESPONSE: See response for figure 3.

COMMENT: Figure 6: suggestion to have all color bars go from white to color and then hatch out the insignificant/no data values. Solid blocks of color are more difficult with color blindness.

RESPONSE: We will make this change

COMMENT: Figure 8: Mention what calibration time period is represented in this figure.

RESPONSE: We will make this change.

COMMENT: Figure 9: Again, suggestion to NOT have green has the zero value, have white.

RESPONSE: We will change the figure to have a color closer to white. We cannot assign white for 0 as it falls in between two groups. But we get the Referee's point and will improve the figure based on the feedback.

COMMENT: Figures in general, increase the size of the text

RESPONSE: We will increase the font size.

COMMENT: Suggested citations: Alexander, M.R., J.K. Pearl, D.A. Bishop, E.R Cook, K.J. Anchukaitis, N. Pederson, The potential to strengthen temperature reconstructions in ecoregions with limited tree line using a multi species approach, Quaternary Research, 1-15, doi: 10.1017/qua.2019.33, 2019 Pearl, J.K., K.J. Anchukaitis, N. Pederson, J. Donnelly, Multivariate climate field reconstructions using tree-rings for the northeastern United States, Journal of Geophysical Research – Atmospheres, doi:10.1029/2019JD031619, 2019

RESPONSE: We will include these citations.

---

## Author Comment (AC2) · 9 Jun 2020

We would like to thank the Referee for the constructive review. The feedback provided will help us to improve the manuscript. Written below are our point by point responses to the Referee's comments. Our responses are below each comment and are the changes we propose to the manuscript based on the Referee's comments. The revised version of the manuscript will be prepared based on the decision of the Editor.

Anonymous Referee #2

COMMENT: The work by Maxwell et al. explores whether an improvement of the tree-ring based Living Blended Drought Atlas (Cook et al., 2010) can be achieved over the Ohio River Valley, US, by increasing the density of the proxy network, as well as incorporating a broader range of tree species. The work also briefly assess whether including the last decades in the calibration/validation exercise might change the performance of the reconstruction models. Overall I find the ideas of this work compelling, and thus I regard the results being of general and international interest. The data is treated with more or less standard methods within the field of dendrochronology and the analyses appear to be sound. Moreover, I find the idea of combining multiple species to obtain a more robust re-construction compelling. Although the study has a sound rationale and execution, the authors' approach appear to have had marginal benefits. In light of this, I miss a discussion on quality versus quantity of predictors in state-of-the-art field reconstructions. Listed below, are a few more specific comments which I hope will help the authors improve the final manuscript.

RESPONSE: Thank you for the general positive response. We would argue that while the patterns are generally the same, the difference in extremes make a big difference in our understanding of past extremes in hydroclimate. We will make this point clearer to ensure readers will understand the importance of our findings. Part of this expansion will consist of broadening our discussion about quantity and quality of gridded reconstructions.

Specific comments: COMMENT: P1/L32: "By sampling tree in 2010 [...] " reword to "By extending the calibration period to 2010 [...]" (suggestion)

RESPONSE: We will make this change.

COMMENT: P2/L50: Oliver et al., 2019 is missing in the reference list. Also, tree-ring based drought reconstruction are not restricted to the mid-latitudes and certainly not only to US (which is somehow implied by the references the authors cite)

RESPONSE: Thank you for catching this, we will add Oliver et al. 2019 to the reference list and ensure we cite a better representation of articles that are beyond the US.

COMMENT: P2/L54: "[...] creating a 2.5âŮęx 2.5âŮęreconstruction" was it not a 2âŮęx

3âŬ̧ePDSI grid that was used in Cook et al., 1999?

RESPONSE: Yes, you are correct. Thank you for catching this, we will make the change.

COMMENT: P2/L55: "The NADA produced multiple centuries of both spatial and temporal data of drought variability" remove either "multiple centuries" or "temporal" – its redundant to have both

RESPONSE: We will remove "temporal"

COMMENT: P3/L81: "Yet, developing a reconstruction assumes that this climate-tree-growth relationship is stationary over time. This assumption was generally true in the early development of the field of dendrochronology (ca. 1920s–1950s; Fritts, 1976). However, as human activities drive the Earth's climate system into historically unprecedented, and potentially non-stationary and non-analogous conditions (Milly et al., 2008), exceptions to this assumption have emerged." Please rephrase, it's unclear and contradictory. By pointing out that the system is stationary between 1920s-1950s the authors also admit that the relationship is non-stationary.

RESPONSE: Thank you, our intention was to highlight when this work was generally being done but we see the confusion now and will reword to improve clarity.

COMMENT: P3/L86: "Changes in the drought signal recorded by tree rings have been established only recently [...]" Do the authors refer to the midwestern US? If so, please be more specific.

RESPONSE: We were referring to a changing signal in drought in general. While there has been a lot of research on changing temperature signals, that is not the case for hydroclimate variability. But yes, the documentation has been recently in the midwestern and eastern US. We will reword to improve clarity.

COMMENT: P4/L97: "...if the year when trees are sampled influences the climate reconstruction" This sentence is awkward, please rephrase. It is not the year when the

trees are sampled, but rather the period that is covered by the calibration period that might influences the reconstruction skill.

RESPONSE: Great point, we will make the change.

COMMENT: P4/L98: "We calibrate the reconstruction with recent (post-1980) radial growth and climate data..." Please rephrase. It is not the reconstruction that is calibrated, but the tree-ring data that is calibrated with climate data to obtain the reconstruction.

RESPONSE: We will reword.

COMMENT: P5/L126 "We used the list method to visually crossdate all samples, and then the pro-gram COFECHA to statistically verify the crossdating" (suggestion)

RESPONSE: We will make the suggested change.

COMMENT: P5/L147: indicate the period for the correlation analysis, and also the significance level applied in this screening

RESPONSE: We will make the suggested change.

COMMENT: P6/149: what is the rational behind using a 250-km search radius? Was it selected based on the spatial characteristics of regional drought climatology observed in the instrumental data?

RESPONSE: We tried a few different radii, but 250-km worked well with the density of our tree-ring network. Basically, the density determined how small the radius could be. If we had an even denser network, we could do a smaller search radius. Similarly, a less dense network would require a higher search radius. We will justify this decision in the resubmission.

COMMENT: P6/L150: it should be mentioned that a dynamic search radius was used to produce LDBA, with the requirement that at least five chronologies had to be located around each grid point. By eyeballing the very sparse tree-ring network in fig 1A I

would assume that the search radius might even had been larger than 450 km across the ORV region. Please check if this is the case, and clarify in the text.

RESPONSE: This is an excellent point and thank you for catching our mistake. We will look into the actual radius of the LBDA for the study region.

COMMENT: P6/L151: did the authors also consider lagged associations between tree-ring and drought data? This has been done in LDBA (i.e. tree-ring data year t + 1 were considered in the reconstruction of drought in year t), meaning that there were actually potentially twice as many predictors of drought at each grid point compared to the number of tree-ring chronologies located around each grid. Also, were there any requirements about the minimum no of chronologies to be included in the predictor pool for the new ORV reconstruction?

RESPONSE: Yes, we did use the t +1 in addition to year t. We will make sure to add the needed text in the resubmission. We did not have a requirement of minimum no of chronologies but used the EPS of 0.85 to determine how far back the reconstructions could go. All gridded recons had more than 5 chronologies in the calibration period. We will add text about this and add a supplemental figure showing the number of chronologies used in the common period for each gridded reconstruction.

COMMENT: P6/L161 and P9/L239, L295: The spectral properties of the resulting hydroclimate reconstructions will be affected by the way the short-lag autocorrelation structure in the tree-ring data is treated. If I am not mistaken, for the LDBA reconstruction a low-order AR model was fitted to both the instrumental and tree-ring series to correct for the mismatch in the short-term autocorrelation. The prewhitened time series were then used to test the association between drought and tree-growth and to build the regression models. The autocorrelation of instrumental data was then added back to the final tree-ring reconstructions of drought. It should be mentioned if a similar approach was adopted also in this study.

RESPONSE: Thank you for this important point. Yes, we followed the exact methods

[Figure]

Interactive
comment

for the LDBA reconstruction. We will be sure to add this information into the text during the revision.

COMMENT: P8/L220: "The ORV reconstructions were shorter in length (maximum of 343 years)compared to the LBDA reconstructions (maximum of 2,006 years) due to each grid reconstruction having a smaller search radius (250 km vs 450 km) for chronology inclusion." This sentence needs to be rephrased. The ORV reconstruction is not shorter because of a smaller search radius, but because the temporal extension of the tree-ring network was more limited than in the LDBA.

RESPONSE: We understand your confusion based on how the sentence was written. We were trying to say that a very old chronology can be used in multiple grids that are quite far away with a larger search radius. The baldcypress in Missouri, for example, allowed many of the gridded reconstructions to go back much further in time for the LBDA. But yes, your point is also true. We will make sure to rephrase to increase clarity.

COMMENT: P9/L248: Not sure I understand how the beta-weight values for the different species were obtained. Are these the loadings from the PCA?

RESPONSE: Yes, thank you for pointing out this confusion. These are indeed the loadings from the PCA. We will edit this text in the resubmission. However, in thinking about this, we will add a 2nd panel that also shows the correlation between each chronology and the instrumental PMDI.

COMMENT: P11/L280 "compared to"

RESPONSE: We will make this change

COMMENT: P11/L283: "multiple gridded reconstructions" perhaps "multiple grid points" would be better suited here

RESPONSE: We will make this change.

COMMENT: P14/361: "[...] calibrating our models " do the authors mean validating? Conclusions: I am missing a sentence or two about future prospects/possibilities of extending the newly sampled data in the ORV region back in time.

RESPONSE: We do mean calibrating because we are not including years up to 2010. But it is the validation statistics that change. We will rephrase to increase clarity.

COMMENT: Figure 1: please add a scale ruler for reference. Also, it might not be clear what the rectangle in the figure represents.

RESPONSE: We will make these changes.

COMMENT: Figure 4: the spatial patterns in the ORV and LBDA reconstructions look pretty similar to me. I would therefore be careful to conclude, based only on this plot (as well as figs1-3 in the supplement), that the ORV reconstruction better match the distribution of soil moisture values and the spatial patterns of the instrumental data compared to the LBDA reconstruction" (L233). The authors need to perform some additional analysis to support this conclusion. For instance, the authors could compute point-by-point correlations between all possible pair of grid points in the instrumental data, ORV and LBDA, respectively, and then plot the correlation as a function of distance between gridpoints (correlation decay distance). If the spatial characteristics of droughts in the ORV reconstruction is indeed more accurate than in the LBDA, then the CDD of the ORV would be more similar to instrumental data. The slope of the correlation vs distance curve would be much less steep for the LBDA reconstruction, because of higher spatial autocorrelation

RESPONSE: It is more of the differences in extreme values rather than the spatial pattern. However, we like your suggestion of the correlation decay distance and will explore it in the resubmission.

COMMENT: Figure 5: the information in this figure loses some of its value if not compared /validated against the spectral properties of the instrumental data. This could be

done by restricting the analysis to the modern period when also instrumental data is available

RESPONSE: Thank you for this point. We will either change the period of analysis or add a panel showing the data between all three during the period of instrumental data.

COMMENT: Figure 6: please indicate in the different figures whether the flips refers to wet, dry or total flips.

RESPONE: We will make this change.

COMMENT: Figure 8: add the periods for calibration and verification either in the figure or in the caption. Also, the figures is not easily interpreted. I suggest the authors add a third column where the differences/residuals between ORV and LBDA calibration and verification statistics are shown.

RESPONSE: We will make these changes.

COMMENT: P7/L200 mentions that the 1941-1980 period was used for validation, while in fig 9 caption it says that calibration period ended 2010. Please clarify. Not all the text is visible in the supplemental Table 1. The timespan of the chronologies should be included. Also, the state abbreviations would probably be meaningless for most of the international readership (at least they should be defined in the caption if the authors decide to keep them)

RESPONSE: This is a comparison of the validation statistics between reconstruction models that ended in 1980 and in 2010 to see how the statistics changed. We will ensure the text in line 200 is clear in the resubmission. Thank you for the suggestions for supplemental table 1. We will make those changes.

COMMENT: There are two figure 3 in the supplement

RESPONSE: Thank you, we will correct the mistake.

---

## Author Response (AR1)

We would like to thank the Referee for the constructive review. The feedback provided helped us to improve the manuscript. Written below are our point by point responses to the Referee's comments. Most of the responses are similar to our initial responses but in doing some of the revisions, a few changes are different from what we initially stated but we make that clear in the point by point responses.

Anonymous Referee #1

COMMENT: "Sampling density and date influence spatial representation of tree ring reconstructions" uses an updated, multi-species, tree-ring network in the Ohio River Valley to demonstrate the influence of increased predictor density and record length on spatial drought reconstructions. This paper presents well-supported findings that increasing predictor density in a gridded hydroclimate reconstruction can identify more localized patterns and emphasizes that the hydrologic sensitivity of some of the species is changing. The authors discuss the influence of recent dampening of extreme droughts and pluvials compared to the 1900-1980 time period, and how incorporating non- traditional dendroclimatology species can strengthen the reconstructions.
The clearly laid out discussion showed the power and limitations or increasing the predictor density in a spatial reconstruction. The authors' conclusions regarding the "fading drought signal" of trees is important for future hydroclimate reconstructions – particularly in this region.
This manuscript should be published, with a few minor edits. A multi species approach to climate reconstructions, comparing the NADA to a denser predictor network in data-scare areas (most recently in Pearl et al., 2019), and adjusting the calibration/verification time period for reconstructions have all been done previously in other regions of the U.S. . Thus, the main novelty of this study is the site geography. The discussion, therefore, would benefit from the inclusion of the author's thoughts on a climatological explanation for the higher number of flips at a local scale compared to large scale. That is, why does the Ohio River Valley experience these flips, how is this distinct from the large scale regional dynamics that "smooth" out these flips in the lower resolution reconstruction?

RESPONSE: Thank you for the positive comments on the manuscript. We included work from Pearl et al., 2019 and have better explained that the greater number of flips is due to a better representation of local variability.

Minor comments:
COMMENT: Be careful when hyphenating "tree ring". It should be "tree-ring" when used as an adjective or modifier, and "tree ring" when use as a noun or direct object. E.g. "tree- ring reconstruction"

RESPONSE: Thank you, checked all uses and ensured we only use a hyphen when grammatically correct.

COMMENT:. . the paper also looks at species not just sample density and length of the record. Perhaps "Sampling characteristics" or "Predictor characteristics influence climate patterns/phenomena in tree-ring reconstructions:

RESPONSE: Great suggestion, we changed the title in the spirit of this comment.

COMMENT: Line 45: add "of climate" after reconstructions so that readers know you are not reconstructing the mechanisms mentioned at the beginnings of the sentence.

RESPONSE: We made this change.

COMMENT: Line 47: delete "historical" – tree rings provide context in the prehistory too.

RESPONSE: We made this change

COMMENT: Line 48: "instrumentally recorded" is awkward. "Droughts and pluvials captured in the instrumental record. . . "or similar

RESPONSE: Good suggestion, we made this change.

COMMENT: Line 77: Pearl et al., 2019 did this in New England

RESPONSE: We added Pearl et al. 2019 to the paper.

COMMENT: Line 274: Again, I think either "historical" or "past" should be here, both are redundant

RESPONSE: We made this change.

COMMENT: Line 295: I would also cite Pearl et al, 2019

RESPONSE: We made this change.

COMMENT: Line 319: Alexander et al 2019 also saw this in a temperature reconstruction

RESPONSE: We added this citation.

COMMENT: Line 275: replace "but" with "be"

RESPONSE: We made this change.

COMMENT: Figure 1: Suggestion to move the USA map and species symbols to the top of the figure. Its odd that its in between panel "A" and "B"

RESPONSE: We made this change.

COMMENT: Figure 3: I would choose either contour lines or un-smoothed squares. Since the maps are not "filled" or "smoothed" the contours are unnecessary and distracting. Reference the color bar in the caption. I also suggest having white (not green or yellow) as 0 PMDI, and then hatch out the grid cells with no data.

RESPONSE: We removed the contours as the reviewer suggested. We changed the colorbar of the figure to have a color closer to white for the two groups that were close to zero. We cannot assign white for 0 as it falls in between two groups. We blacked out cells with no data.

COMMENT: Figure 4: same comment as 3

RESPONSE: See response for figure 3.

COMMENT: Figure 6: suggestion to have all color bars go from white to color and then hatch out the insignificant/no data values. Solid blocks of color are more difficult with color blindness.

RESPONSE: We made these changes and blacked out the no data values.

COMMENT: Figure 8: Mention what calibration time period is represented in this figure.

RESPONSE: We made this change.

COMMENT: Figure 9: Again, suggestion to NOT have green has the zero value, have white.

RESPONSE: We changed the colorbar of the figure to have a color closer to white for the two groups that were close to zero. We cannot assign white for 0 as it falls in between two groups. We blacked out cells with no data.

COMMENT: Figures in general, increase the size of the text

RESPONSE: We made this change.

COMMENT: Suggested citations:
Alexander, M.R., J.K. Pearl, D.A. Bishop, E.R Cook, K.J. Anchukaitis, N. Pederson, The potential to strengthen temperature reconstructions in ecoregions with limited tree line using a multi species approach, Quaternary Research, 1-15, doi: 10.1017/qua.2019.33, 2019
Pearl, J.K., K.J. Anchukaitis, N. Pederson, J. Donnelly, Multivariate climate field reconstructions using tree-rings for the northeastern United States, Journal of Geophysical Research – Atmospheres, doi:10.1029/2019JD031619, 2019

RESPONSE: We included citations.

We would like to thank the Referee for the constructive review. The feedback provided helped us to improve the manuscript. Written below are our point by point responses to the Referee's comments. Most of the responses are similar to our initial responses but in doing some of the revisions, a few changes are different from what we initially stated but we make that clear in the point by point responses.

Anonymous Referee #2

COMMENT: The work by Maxwell et al. explores whether an improvement of the tree-ring based Living Blended Drought Atlas (Cook et al., 2010) can be achieved over the Ohio River Valley, US, by increasing the density of the proxy network, as well as incorporating a broader range of tree species. The work also briefly assess whether including the last decades in the calibration/validation exercise might change the performance of the reconstruction models.
Overall I find the ideas of this work compelling, and thus I regard the results being of general and international interest. The data is treated with more or less standard methods within the field of dendrochronology and the analyses appear to be sound. Moreover, I find the idea of combining multiple species to obtain a more robust re-construction compelling. Although the study has a sound rationale and execution, the authors' approach appear to have had marginal benefits. In light of this, I miss a discussion on quality versus quantity of predictors in state-of-the-art field reconstructions. Listed below, are a few more specific comments which I hope will help the authors improve the final manuscript.

RESPONSE: Thank you for the general positive response. We argue that while the patterns are generally the same, the difference in extremes make a big difference in our understanding of past extremes in hydroclimate. We made this point clearer to ensure readers will understand the importance of our findings. We also expanded the discussion about quantity and quality of gridded reconstructions.

Specific comments:
COMMENT: P1/L32: "By sampling tree in 2010 [...] " reword to "By extending the calibration period to 2010 [...]" (suggestion)

RESPONSE: We made this change.

COMMENT: P2/L50: Oliver et al., 2019 is missing in the reference list. Also, tree-ring based drought reconstruction are not restricted to the mid-latitudes and certainly not only to US (which is somehow implied by the references the authors cite)

RESPONSE: Thank you for catching this, we added Oliver et al. 2019 to the reference list and added a better representation of articles that are beyond the US as well.

COMMENT: P2/L54: "[...] creating a 2.5°x 2.5°reconstruction" was it not a 2°x 3°PDSI grid that was used in Cook et al., 1999?

RESPONSE: Yes, you are correct. Thank you for catching this, we made the change.

COMMENT: P2/L55: "The NADA produced multiple centuries of both spatial and temporal data of drought variability" remove either "multiple centuries" or "temporal" – its redundant to have both

RESPONSE: We removed "temporal"

COMMENT: P3/L81: "Yet, developing a reconstruction assumes that this climate-tree-growth relationship is stationary over time. This assumption was generally true in the

early development of the field of dendrochronology (ca. 1920s–1950s; Fritts, 1976). However, as human activities drive the Earth's climate system into historically unprecedented, and potentially non-stationary and non-analogous conditions (Milly et al., 2008), exceptions to this assumption have emerged." Please rephrase, it's unclear and contradictory. By pointing out that the system is stationary between 1920s-1950s the authors also admit that the relationship is non-stationary.

RESPONSE: Thank you, our intention was to highlight when this work was generally being done but we see the confusion now and have removed it to improve clarity.

COMMENT: P3/L86: "Changes in the drought signal recorded by tree rings have been established only recently [...]" Do the authors refer to the midwestern US? If so, please be more specific.

RESPONSE: We were referring to a changing signal in drought in general. While there has been a lot of research on changing temperature signals, that is not the case for hydroclimate variability. But yes, the documentation has been recently in the midwestern and eastern US.

COMMENT: P4/L97: "...if the year when trees are sampled influences the climate reconstruction" This sentence is awkward, please rephrase. It is not the year when the trees are sampled, but rather the period that is covered by the calibration period that might influences the reconstruction skill.

RESPONSE: Great point, we made the change.

COMMENT: P4/L98: "We calibrate the reconstruction with recent (post-1980) radial growth and climate data..." Please rephrase. It is not the reconstruction that is calibrated, but the tree-ring data that is calibrated with climate data to obtain the reconstruction.

RESPONSE: We rephrased this sentence and other areas of the manuscript that had a similar problem.

COMMENT: P5/L126 "We used the list method to visually crossdate all samples, and then the pro-gram COFECHA to statistically verify the crossdating" (suggestion)

RESPONSE: We made suggested change.

COMMENT: P5/L147: indicate the period for the correlation analysis, and also the significance level applied in this screening

RESPONSE: We made suggested change.

COMMENT: P6/149: what is the rational behind using a 250-km search radius? Was it selected based on the spatial characteristics of regional drought climatology observed in the instrumental data?

RESPONSE: We tried a few different radii, but 250-km worked well with the density of our tree-ring network. Basically, the density determined how small the radius could be. If we had an even denser network, we could do a smaller search radius. Similarly, a less dense network would require a higher search radius. We added to text to justify this decision in the resubmission, mainly that this was the radius that still allowed five chronologies to be included in each gridded reconstruction.

COMMENT: P6/L150: it should be mentioned that a dynamic search radius was used to produce LDBA, with the requirement that at least five chronologies had to be located around each grid point. By eyeballing the very sparse tree-ring network in fig 1A I

would assume that the search radius might even had been larger than 450 km across the ORV region. Please check if this is the case, and clarify in the text.

RESPONSE: This is an excellent point and thank you for catching our mistake. We now discuss that the LBDA could have a larger radius than 450km in sparse regions.

COMMENT: P6/L151: did the authors also consider lagged associations between tree-ring and drought data? This has been done in LDBA (i.e. tree-ring data year t + 1 were considered in the reconstruction of drought in year t), meaning that there were actually potentially twice as many predictors of drought at each grid point compared to the number of tree-ring chronologies located around each grid. Also, were there any requirements about the minimum no of chronologies to be included in the predictor pool for the new ORV reconstruction?

RESPONSE: Yes, we did use the t +1 in addition to year t. We have added the needed text in the resubmission. We also used the five chronology limit for the gridded recons (we mistakenly said that was not the case in the initial reply). All gridded recons had at least 5 chronologies in the calibration period. We added text about this and added a supplemental figure showing the number of chronologies used in the common period for each gridded reconstruction.

COMMENT: P6/L161 and P9/L239, L295: The spectral properties of the resulting hydroclimate reconstructions will be affected by the way the short-lag autocorrelation structure in the tree-ring data is treated. If I am not mistaken, for the LDBA reconstruction a low-order AR model was fitted to both the instrumental and tree-ring series to correct for the mismatch in the short-term autocorrelation. The prewhitened time series were then used to test the association between drought and tree-growth and to build the regression models. The autocorrelation of instrumental data was then added back to the final tree-ring reconstructions of drought. It should be mentioned if a similar approach was adopted also in this study.

RESPONSE: Thank you for this important point. Yes, we followed the exact methods for the LDBA reconstruction. We added this information into the text during the revision.

COMMENT:  P8/L220: "The ORV reconstructions were shorter in length (maximum of 343 years)compared to the LBDA reconstructions (maximum of 2,006 years) due to each grid reconstruction having a smaller search radius (250 km vs 450 km) for chronology inclusion." This sentence needs to be rephrased. The ORV reconstruction is not shorter because of a smaller search radius, but because the temporal extension of the tree-ring network was more limited than in the LBDA.

RESPONSE: We understand your confusion based on how the sentence was written. We were trying to say that a very old chronology can be used in multiple grids that are quite far away with a larger search radius. The baldcypress in Missouri, for example, allowed many of the gridded reconstructions to go back much further in time for the LBDA. But yes, your point is also true. We have rephrased to increase clarity.

COMMENT: P9/L248: Not sure I understand how the beta-weight values for the different species were obtained. Are these the loadings from the PCA?

RESPONSE: Yes, thank you for pointing out this confusion. These were indeed the loadings from the PCA. In thinking through this we have changed the figure to correlation coefficient of the species to the instrumental PMDI for the gridded reconstructions. This better represents what we were trying to show and did require some minor text changes.

COMMENT: P11/L280 "compared to"

RESPONSE: We made this change

COMMENT: P11/L283: "multiple gridded reconstructions" perhaps "multiple grid points" would be better suited here

RESPONSE: We made change.

COMMENT: P14/361: "[...] calibrating our models " do the authors mean validating? Conclusions: I am missing a sentence or two about future prospects/possibilities of extending the newly sampled data in the ORV region back in time.

RESPONSE: We do mean calibrating because we are not including years up to 2010. But it is the validation statistics that changed. We have rephrased this to increase clarity.

COMMENT: Figure 1: please add a scale ruler for reference. Also, it might not be clear what the rectangle in the figure represents.

RESPONSE: We made these changes.

COMMENT: Figure 4: the spatial patterns in the ORV and LBDA reconstructions look pretty similar to me. I would therefore be careful to conclude, based only on this plot (as well as figs1-3 in the supplement), that the ORV reconstruction better match the distribution of soil moisture values and the spatial patterns of the instrumental data compared to the LBDA reconstruction" (L233). The authors need to perform some additional analysis to support this conclusion. For instance, the authors could compute point-by-point correlations between all possible pair of grid points in the instrumental data, ORV and LBDA, respectively, and then plot the correlation as a function of distance between gridpoints (correlation decay distance). If the spatial characteristics of droughts in the ORV reconstruction is indeed more accurate than in the LBDA, then the CDD of the ORV would be more similar to instrumental data. The slope of the correlation vs distance curve would be much less steep for the LBDA reconstruction, because of higher spatial autocorrelation

RESPONSE: This was a great suggestion and the results supported the other analyses nicely. Thank you for the suggestion, we have added it as the new figure 5 in the manuscript and added the appropriate text in the methods, results, and discussion.

COMMENT: Figure 5: the information in this figure loses some of its value if not compared /validated against the spectral properties of the instrumental data. This could be done by restricting the analysis to the modern period when also instrumental data is available

RESPONSE: Thank you for this point. We changed the period of analysis to the instrumental record and added the instrumental data to the figure. Again, this supported our previous findings, thank you.

COMMENT: Figure 6: please indicate in the different figures whether the flips refers to wet, dry or total flips.

RESPONE: We made this change.

COMMENT: Figure 8: add the periods for calibration and verification either in the figure or in the caption. Also, the figures is not easily interpreted. I suggest the authors add a third column where the differences/residuals between ORV and LBDA calibration and verification statistics are shown.

RESPONSE: We made these changes.

COMMENT: P7/L200 mentions that the 1941-1980 period was used for validation, while in fig 9 caption it says that calibration period ended 2010. Please clarify. Not all the text is visible in the supplemental Table 1. The timespan of the chronologies should be included. Also, the state abbreviations would probably be meaningless for most of the international readership (at least they should be defined in the caption if the authors decide to keep them)

RESPONSE: This is a comparison of the validation statistics between reconstruction models that ended in 1980 and in 2010 to see how the statistics changed. We have made this easier to interpret in the resubmission. Thank you for the suggestions for supplemental table 1. We made those changes.

COMMENT: There are two figure 3 in the supplement

RESPONSE: Thank you, we corrected the mistake.

[revised manuscript text omitted]

---

## Author Response (AR2)

Editor Comments
Thank you for responding so well to the comments in the interactive discussion. As you have seen both reviewers are positive to your work, but suggest some additional work to make it more accessible. I thus invite you to revise your manuscript in the line with the comments provided and your responses. When I have received the new version, I will send it out for a new round of review.

Thank you for your help and constructive feedback for this manuscript. Below, we have responded to each point and denote the changes in the resubmitted (R2) manuscript.

"Sampling density and date along with species selection influence spatial representation of tree-ring reconstructions" is a clear, concise manuscript comparing a gridded drought construction made from network of updated, multi-species, tree-ring chronologies in the Ohio River Valley with the North American Drought Atlas. This comparison demonstrates the influence of increased predictor density and record length on gridded drought reconstructions.
The clearly laid out discussion showed the power and limitations or increasing the predictor density in a spatial reconstruction. The authors' conclusions regarding the "fading drought signal" of trees is important for future hydroclimate reconstructions – particularly in this region. This manuscript should be published pending minor edits, please see the below line items.

Thank you for the constructive feedback that improved this manuscript.

Line 22: suggestion to replace "are represented" by "are modeled by few tree-ring chronologies"
Made suggested change

Line 45: replace "represent" with "are"
Made suggested change

Line 46: replace "/" with "and"
Made suggested change

Line 55: delete "an" and replace "to extreme" with "for extreme"
Made suggested change

Line 61: Suggestion "They are most useful for large, regional (what is regional here? maybe instead of regional be specific on what scale of climate/ecosystem ranges/ grid cells etc.) events"
Changed to sub-continental

Line 61-62: Instead of "each gridded reconstruction" "The reconstruction at each grid cell uses tree-ring data that are within a 450-km radius of that grid point"
Made suggested change

Line 62: Rather than "therefore", I suggest more specificity: by pulling from such a wide range of predictors, the NADA and LBDA models excel at representing large-scale phenomena as they tend to average out smaller scale features…
Made suggested change

Line 77: perhaps elaborate on why the chronologies not being updated is important.
Added some expansion text.

Line 79-83: Between this sentence and the preceding paragraph it feels a bit redundant
We would argue that this sentence sets up the next paragraph and did not make a change.

Line 96: add "despite a fading drought signal" at the end "of the instrumental data"
Made suggested change.

Line 98: you're referencing the reconstructions with the new, dense, network right? Possibly specify which reconstructions you're discussing.
Thank you, we added clarification.

Line 113: "significantly correlated" at what alpha level? Cook 2010 does 0.01 I believe?
added p-value

Line 127: You mention "standard field methods" --- standard sampling design, or standard collection methods - from dendroclimatology (canopy dominant oldest) rather than ecology? Coring the tree at breast height seems more ecology to me
Thank you, however, here are hundreds of dendroclimate articles that core trees at breast height. Therefore, we left this as is.

Line 142: regarding signal free standardization, possibly replace "standardization reduces" with "standardization can reduce". Just to place my arbitrary horse into a stupid race – Signal Free isn't a silver bullet to reduce the trend distortion- I'm glad it does here for this network.
Agreed, def. not the silver bullet, made suggested change.

Line 145: EPS value of 0.80 – why not 0.85 as in the reference cited (Wigley et al., 1984)
The Wigley et al. article states you can use several different cutoffs but gave the 0.85 as an example. Several studies have used 0.80 as well. Here, we did that because a few of the crns were able to go back farther and were just slightly below the 0.85 level. The cutoff is pretty subjective and we just wanted to be open about it.

Line 152: Only the chronologies with significant at $p < 0.05$ level – is this the same as Cook, 2010? Please state if you are as, more, or less stringent in your cut off
Looking into the Cook et al. 2010, it seems they used 0.1, so this would be more stringent.

Line 164: replace "conducting" with "conducted"
Made suggested change

Line 166: You have already defined JJA, don't need to redefine
Thanks, corrected.

Line 168: as done by Cook 2010?
Yep, we tried to replicate as much as possible.

Line 176-Line177: You use simple Pearson's correlation rather than beta weights of the species to the recon – why?

As we stated in the previous reviewer remark response, the beta weights of the model will be for the PCs retained and not allow us to examine the impact of the species crn. We initially had the beta weights on how well each species loaded to the PCs, but this was an indirect measure of species contribution so we decided to present the correlation as a more direct comparison.

Line 196: Again, why 0.8 not 0.85?

See response to line 145.

Line 218-220: This is really neat. One of the reasons I think this paper is great!

Thanks!

Line 227: My suggestion to just say "whereas 18 of the sites included…" rather than "n=18"

Good suggestion! Made the change.

Line 227: Do you mean multi-species when you say multiple chronology?

Yes, nice catch. Made the change.

Line 238: really a maximum of 2006 years in this location as well? I'm surprised - I don't think it goes back much past 490CE in this region as far as I can tell from the raw data files. If you mean 2006 years over the entire network, better to specify.

Good catch, the maximum was actually 1,645 years in the region for the LBDA. We have made the change.

Line 248: I would put the sentence starting with "Specifically" in the methods.

Good point, we moved to methods.

Line 386" "suggesting a wet period"… very very cool.

Thanks!

Figures: in general- the figures are much improved
Figure 2: suggestion for consistency put degree N and degree W on the grid

Thanks for the suggestion, instead, we changed the labels on Figure 1 to match the other figures.

Figure 3: Make PMDI bigger

Made the change

Figure 4: It would be better for the readers to contextualize this figure a bit more with a brief sentence on the climatology of 1954 in the maintext- we can see from the instrumental that we have wide spread drought in much of the area but slightly wetter conditions in the north – is there a reference about this particular summer?

We understand your point, but we also compare other years besides just 1954. We have added references in the methods section to allow readers to explore these events in more detail if they like.

Figure 7: increase the size of "number of flips" and make the font the same kind as the sup titles?
Made suggested changes

Figure 9: I suggest to put a color bar for the calibration R2 as well since every other figure has a color bar?
Thanks for the suggestion, because it would be identical to the verification colorbar, we decided to not add a colorbar here.

Figure 10: Again, ensure that the font is the same and increase the color bar label and color bar axes. Also I suggest to keep consistent with your figure formats: choose for your figures degree N and W (like figure 1) or "lat and lon" (like the rest of them).
Made suggested changes

[revised manuscript text omitted]